# LoRA-EnVar: Parameter-Efficient Hybrid Ensemble Variational Assimilation for Weather Forecasting

**Yi Xiao**
Tsinghua University, Beijing
Shanghai Artificial Intelligence Laboratory, Shanghai
`y-xiao22@mails.tsinghua.edu.cn`

**Hang Fan**
Columbia University, New York
`hf2526@columbia.edu`

**Kun Chen**
Fudan University, Shanghai
Shanghai Artificial Intelligence Laboratory, Shanghai
`kunc3301@163.com`

**Ye Cao**
Tsinghua University, Beijing
`caoye541@gmail.com`

**Ben Fei**[✉]
The Chinese University of Hong Kong, Hong Kong
Shanghai Artificial Intelligence Laboratory, Shanghai
`benfei@cuhk.edu.hk`

**Wei Xue**[✉]
Tsinghua University, Beijing
`xuewei@tsinghua.edu.cn`

**Lei Bai**
Shanghai Artificial Intelligence Laboratory, Shanghai
`baisanshi@gmail.com`

## Abstract

Accurate estimation of background error (i.e., forecast error) distribution is critical for effective data assimilation (DA) in numerical weather prediction (NWP). In state-of-the-art operational DA systems, it is common to account for the temporal evolution of background errors by employing hybrid methods, which blend a static climatological covariance with a flow-dependent ensemble-derived component. While effective to some extent, these methods typically assume Gaussian-distributed errors and rely heavily on hand-crafted covariance structures and domain expertise, limiting their ability to capture the complex, non-Gaussian nature of atmospheric dynamics. In this work, we propose LoRA-EnVar, a novel hybrid ensemble variational DA algorithm that integrates low-rank adaptation (LoRA) into a deep generative modeling framework. We first learn a climatological background error distribution using a variational autoencoder (VAE) trained on historical data. To incorporate flow-dependent uncertainty, we introduce LoRA modules that efficiently adapt the learned distribution in response to flow-dependent ensemble perturbations. Our approach supports online finetuning, enabling dynamic updates of the background error distribution without catastrophic forgetting. We validate LoRA-EnVar in high-resolution assimilation settings using the FengWu forecast model and simulated observations from ERA5 reanalysis. Experimental results show that LoRA-EnVar significantly improves assimilation accuracy over models assuming static background error distribution and achieves comparable or better performance than full finetuning while reducing the number of trainable parameters by three orders of magnitude. This demonstrates the potential of parameter-efficient adaptation for scalable, non-Gaussian DA in operational meteorology.

39th Conference on Neural Information Processing Systems (NeurIPS 2025).

# 1 Introduction

Data assimilation (DA) plays a crucial role in numerical weather prediction (NWP) by integrating real-world observations with model forecasts to produce accurate initial states for future predictions [1, 2, 3, 4]. In this process, a prior forecast, typically generated by a numerical model, is treated as the background state. The effectiveness of DA depends largely on accurately characterizing this background error distribution, which reflects the uncertainty in prior forecasts and determines how new observations are assimilated into the model.

Early DA systems such as 3DVar typically assume a static Gaussian background error, using a fixed covariance matrix throughout the forecast cycle [5, 6, 7, 8]. While computationally efficient, this assumption fails to capture the fact that background errors evolve over time, especially in dynamically active regions [9]. To address this, ensemble-based methods like the Ensemble Kalman Filter (EnKF) estimate flow-dependent covariances from short-term ensemble forecasts [10, 11, 12]. However, in high-dimensional systems like NWP, limited ensemble size often leads to spurious correlations and noisy error estimates [13, 12]. To mitigate this, hybrid methods have become the dominant approach in modern operational DA. These methods combine a static climatological covariance estimated from historical simulations with a flow-dependent covariance derived from real-time ensembles, leveraging the strength of both [9, 14]. Despite their effectiveness, hybrid methods still rely on a Gaussian error assumption [15, 16, 17] and involve expert-crafted covariance structures [18], which may limit their ability to model complex, non-Gaussian background uncertainty [19].

Recent advances in AI-based DA methods aim to overcome the limitations of traditional methods. For example, DiffDA [20] and Score-based DA (SDA) [21, 22] utilize diffusion models to model complex non-Gaussian uncertainty, but they do not incorporate climatological priors or flow-dependent ensembles. Other approaches, such as FuXi-En4DVar [23], introduce ensembles to model flow-dependent uncertainty, yet still rely on Gaussian assumptions. Similarly, the Ensemble Score Filter (EnSF) [24, 25] incorporates flow dependency but scales poorly to high-dimensional systems. Latent-EnSF [26, 27] mitigates this issue by learning a latent representation of error structure, but it lacks a mechanism to incorporate static background statistics, which can limit its long-term stability and climatological consistency.

To date, no existing method has successfully captured **hybrid background error characteristics**, combining both static climatological and flow-dependent features, within **a non-Gaussian generative modeling framework**. This leaves a gap in current methodologies for flexibly and accurately representing the full spectrum of uncertainty in real-world atmospheric data assimilation.

In this work, we propose LoRA-EnVar, a hybrid variational assimilation framework that combines deep generative modeling with parameter-efficient adaptation to capture both climatological priors and flow-dependent background error dynamics, without relying on full retraining or manual covariance design. We first train a variational autoencoder (VAE) [28] on historical forecast error samples to learn a low-dimensional latent representation of climatological background uncertainty. To incorporate real-time flow-dependent information, we introduce low-rank adaptation (LoRA) [29] modules into the decoder of the VAE. During each assimilation cycle, these LoRA modules are finetuned online using perturbations from ensemble forecasts, enabling the generative model to adapt dynamically to the evolving atmospheric states. We validate LoRA-EnVar in high-resolution cyclic assimilation experiments using the FengWu [30] forecast model and ERA5-simulated observations, demonstrating its effectiveness in large-scale NWP settings. Compared to full decoder finetuning, our approach maintains stability, avoids catastrophic forgetting, and reduces the number of trainable parameters by over three orders of magnitude. Our main contributions are summarized as follows:

- We design a hybrid deep generative framework that unifies VAE-based climatological modeling with LoRA-based flow-dependent adaptation.

- We introduce online low-rank finetuning during assimilation cycles to enable dynamic, non-Gaussian background error updates with minimal computational cost.

- We demonstrate consistent accuracy improvements over static and hybrid baselines in high-resolution cyclic NWP experiments.

## 2 Preliminaries

**Variational Assimilation**  In variational data assimilation, the goal is to infer the most likely system state $\mathbf{x}_a$ given a background estimate $\mathbf{x}_b$ and a set of observations $\mathbf{y}$, by maximizing the conditional probability distribution $p(\mathbf{x}|\mathbf{x}_b, \mathbf{y})$. This yields the analysis state $\mathbf{x}_a = \arg\max_{\mathbf{x}} p(\mathbf{x}|\mathbf{x}_b, \mathbf{y})$.

Since the background state $\mathbf{x}_b$ originates from a numerical forecast and the observations $\mathbf{y}$ are obtained from satellites or ground-based instruments, it is reasonable to assume independence between $\mathbf{x}_b$ and $\mathbf{y}$ [3]. Under this assumption, the posterior distribution can be factorized as [19]:

$$\arg\max_{\mathbf{x}} p(\mathbf{x}|\mathbf{x}_b, \mathbf{y}) = \arg\max_{\mathbf{x}} p(\mathbf{y}|\mathbf{x})p(\mathbf{x}|\mathbf{x}_b). \tag{1}$$

This formulation leads naturally to the definition of the variational cost function:

$$\mathcal{L}(\mathbf{x}) = -\log p(\mathbf{y}|\mathbf{x})p(\mathbf{x}|\mathbf{x}_b) = -\log p(\mathbf{y}|\mathbf{x}) - \log p(\mathbf{x}|\mathbf{x}_b). \tag{2}$$

The total cost comprises two components: an observation term $\mathcal{L}_o(\mathbf{x}, \mathbf{y}) = -\log p(\mathbf{y}|\mathbf{x})$ and a background term $\mathcal{L}_b(\mathbf{x}, \mathbf{x}_b) = -\log p(\mathbf{x}|\mathbf{x}_b)$.

The observation term primarily reflects the statistical characteristics of measurement errors and is closely tied to the properties of the observing instruments and data quality. In contrast, the background term captures the uncertainty in the model forecast and the structure of prior errors, which are often more complex and system-dependent. This work focuses on improving the modeling of the background term to enhance the assimilation accuracy.

**Hybrid Background Term Modeling**  The background term models the uncertainty in the prior forecast and plays a critical role in variational data assimilation. Its estimation relies on representative error samples that capture the statistical characteristics of forecast errors. These samples typically fall into two categories:

- Climatological samples, obtained from long-term historical analysis data (typically using NMC method [17]), represent relatively stable and time-invariant error structures that reflect the background error characteristics of the climate state.
- Flow-dependent samples, extracted from short-range ensemble forecasts at the current time, capture transient, situation-specific features of forecast uncertainty.

Traditional variational methods are based solely on climatological samples, while ensemble-based methods use flow-dependent perturbations. To take advantage of both, hybrid methods that combine static and dynamic information have been developed, which have been shown to have better analysis quality and higher assimilation accuracy.

**Traditional Hybrid EnVar**  Traditional data assimilation methods assume that the background error follows a Gaussian distribution, that is, $\mathbf{x} - \mathbf{x}_b \sim \mathcal{N}(\mathbf{0}, \mathbf{B})$, where $\mathbf{B}$ is the background error covariance matrix. In Hybrid EnVar, $\mathbf{B}$ is assumed to be a linear combination of $\mathbf{B}_s$ and $\mathbf{B}_e$, that is

$$\mathbf{B} = \alpha\mathbf{B}_s + (1 - \alpha)\mathbf{B}_e, \tag{3}$$

where $\mathbf{B}_s$ and $\mathbf{B}_e$ denote covariances calculated from the static climatological and ensemble-derived flow-dependent samples, respectively, and $\alpha \in [0, 1]$ is a tunable weighting coefficient. Despite their practical success, most hybrid methods still rely on the Gaussian assumption and linear formulations of $\mathbf{B}$, limiting their expressiveness in highly nonlinear or non-Gaussian regimes. In this work, we retain the core design principle of combining climatological and flow-dependent information, but replace the hand-crafted Gaussian formulation with a deep, adaptive generative model that can learn and update complex background error distributions more expressively.

## 3 LoRA-EnVar

### 3.1 Non-Gaussian Hybrid Background Modeling via Online Adaptation

In traditional variational assimilation, the background error is often assumed to follow a Gaussian distribution, allowing its uncertainty structure to be fully characterized by a covariance matrix $\mathbf{B}$. This makes it feasible to represent $\mathbf{B}$ as a linear combination of static and flow-dependent components. However, in non-Gaussian settings, the background error distribution may exhibit skewness,

multimodality, or higher-order dependencies that a second-order statistic like a covariance matrix cannot capture. In such cases, the distribution must be modeled more expressively, typically through deep generative models such as VAEs [28], normalizing flows [31, 32], or diffusion models [33, 34], which represent complex distributions in a latent or implicit form. Crucially, these models no longer expose an explicit covariance matrix structure, making it nontrivial to incorporate flow-dependent ensemble samples via simple linear blending, as is done in traditional hybrid methods.

To overcome this limitation, we draw on the concept of online learning in neural networks, where a model pre-trained on historical data can be incrementally updated with new samples. This is analogous to the DA scenario, in which long-term historical samples provide a stable prior, while flow-dependent samples, available during each assimilation cycle, reflect evolving dynamics that require continuous adaptation.

Specifically, we build on the VAE-Var framework to model the background error distribution as a deep latent-variable generative model. The VAE is first pre-trained using climatological error samples derived from long-term reanalysis data. This training phase captures the relatively stable structure of background uncertainty.

During assimilation, flow-dependent error samples, which are generated from ensemble forecasts around the current state, become available at each cycle. Rather than discarding prior knowledge and retraining the model from scratch, we treat these new samples as streaming data and update the model incrementally. This approach allows the generative model to evolve over time, incorporating both global climatological priors and localized, transient dynamics.

### 3.2 Parameter-Efficient Online Adaptation with LoRA

While online adaptation provides a principled framework for incorporating flow-dependent information into the background error model, a key challenge lies in how to update the generative model efficiently and stably during assimilation cycles. Directly finetuning the full set of parameters in the VAE decoder for every new assimilation window is computationally expensive and risks overfitting to transient flow features, potentially leading to catastrophic forgetting of the climatological prior learned during pretraining.

To address these issues, we adopt LoRA as a lightweight, parameter-efficient mechanism for online adaptation. LoRA inserts low-rank trainable matrices into the linear layers of the decoder while keeping the original weights frozen. During online updates, only the LoRA parameters are tuned using flow-dependent samples, while the backbone decoder, trained on climatological data, remains unchanged.

This design offers several advantages:

- **Efficiency**: The number of trainable parameters is significantly reduced (approximately 0.1% of the full model), allowing fast updates even in high-resolution settings.
- **Stability**: By preserving the original decoder weights, the model retains its climatological prior knowledge, mitigating the risk of catastrophic forgetting.
- **Flexibility**: Despite its lightweight nature, LoRA introduces enough capacity to fit flow-dependent corrections effectively, enabling the model to adjust its generative outputs to evolving error structures.

By combining LoRA with the online learning structure introduced in Section 3.1, LoRA-EnVar achieves a scalable and dynamically adaptable representation of background error distributions, balancing long-term stability with short-term responsiveness.

### 3.3 LoRA-EnVar Pipeline

The overall architecture of LoRA-EnVar is illustrated in Figure 1. The framework consists of two phases: offline learning and online adaptation.

In the offline phase, we first train a VAE using historical background error samples constructed from long-term historical datasets. This allows the decoder $\mathcal{D}$ of the VAE to model the climatological distribution of background error, capturing large-scale and relatively stable uncertainty patterns.

Once the offline training is complete, the system enters the online phase, where LoRA modules are introduced into the decoder to enable dynamic adaptation to flow-dependent characteristics.

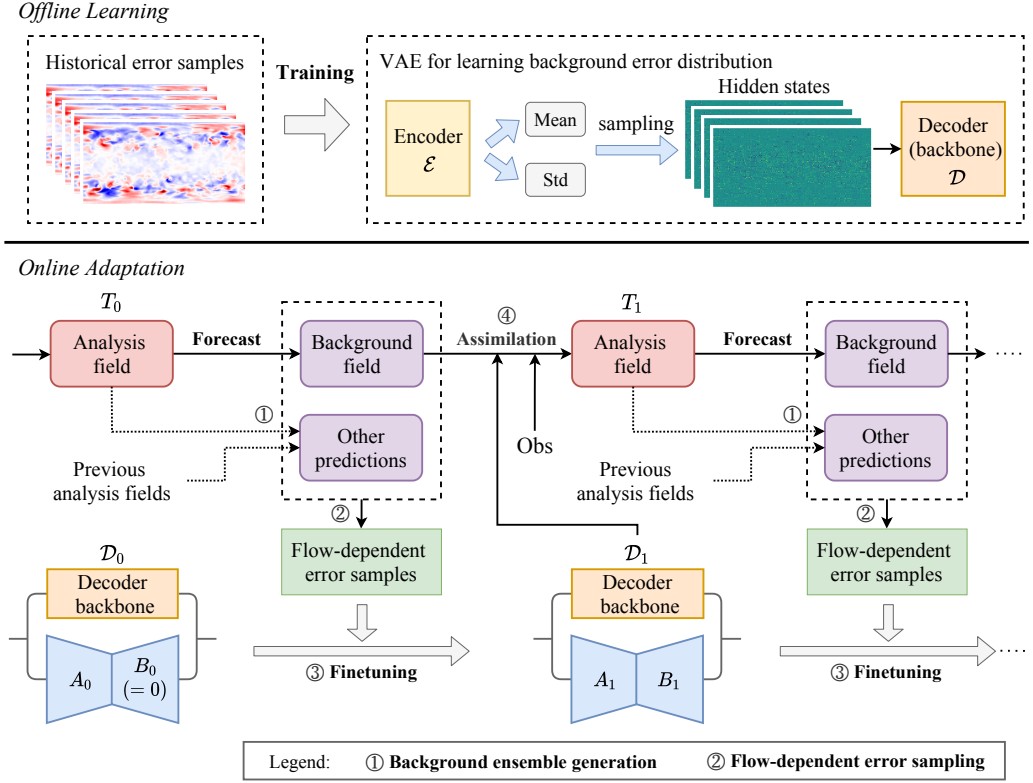

Figure 1: **Overview of the LoRA-EnVar framework.** The method consists of an offline VAE training phase using climatological samples and an online adaptation phase using flow-dependent samples for LoRA finetuning. See Section 3.3 for details.

Following the initialization strategy proposed in the original LoRA paper [29], the low-rank matrices $B_0$ are initialized to zero to preserve the pretrained decoder's output at the beginning of adaptation.

At each assimilation window, we perform the following steps: (1) **Background ensemble generation.** The previous analysis states are used to produce an ensemble of background forecasts for the current time step. This ensemble can be constructed using established techniques, such as time-lagging [35, 36] or breeding methods [37, 38]. (2) **Flow-dependent error sampling.** From the ensemble, we derive perturbations representing the flow-dependent background error samples at the current time. (3) **Online finetuning.** These samples are then used to finetune only the LoRA modules in the VAE, while keeping the backbone parameters frozen. (4) **Assimilation.** The adapted decoder is then used within the variational assimilation framework to compute the analysis field for the current time step. Specifically, we adopt the loss formulation (see Appendix) from VAE-Var, which integrates both the generative model and observation likelihood into a unified variational objective.

This process is repeated at each assimilation window, forming a continuous DA cycle that incrementally updates the background error distribution and generates the analysis fields.

## 4 Results

### 4.1 Experimental Setup

**Forecasting Model**    The experiments in this study are conducted within a real-world global medium-range weather forecasting context. Specifically, we evaluate our data assimilation framework using FengWu [30]. FengWu is a learning-based medium-range weather forecasting model trained on the ERA5 reanalysis dataset. It produces six-hour forecasts and simulates a total of 69 meteorological variables, including five upper-air variables across 13 pressure levels and four surface variables. The

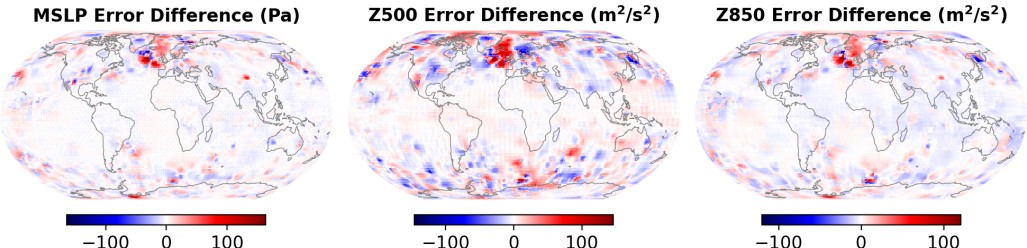

Figure 2: **Spatial error difference between assimilation with and without LoRA finetuning.** Red areas indicate regions where LoRA-EnVar reduces forecast error relative to the frozen VAE baseline. Results are shown for three representative variables, all evaluated at 00:00 UTC on January 1st, 2022. Notable improvements are observed over the northeastern Atlantic, particularly west of the UK.

upper-air variables include geopotential height (z), specific humidity (q), zonal wind (u), meridional wind (v), and air temperature (t), denoted using standard short-name and pressure-level conventions (e.g., z500 refers to geopotential at $500\,\mathrm{hPa}$). The four surface variables are 2-meter temperature (t2m), 10-meter zonal wind (u10), 10-meter meridional wind (v10), and mean sea level pressure (mslp). In our experiments, all simulations are performed at a global resolution of $0.25°$.

**LoRA Module Design** The architecture of the VAE follows the design introduced in VAE-Var [19], where both the encoder and the decoder are built upon the Swin Transformer V2 [39] backbone, which is well-suited to structured meteorological fields and high-resolution global inputs [39, 40]. To enable parameter-efficient online adaptation, we incorporate LoRA modules into the VAE decoder. Following the principle of minimal parameter overhead, we insert LoRA modules only into the query projection (q) layers of each self-attention block in the decoder [41, 32]. This selective design preserves the original network's structure while allowing low-rank corrections to flow-dependent background error features. Unless otherwise stated, the LoRA rank is fixed at 2 across all experiments (see the Appendix for ablation studies).

## 4.2 Single-Step Assimilation Experiment

We begin by evaluating the effectiveness of LoRA-EnVar's hybrid modeling ability in a single-step assimilation setting. Specifically, we conduct assimilation experiments targeting 00:00 UTC on January 1st, 2022. The background state is obtained by running FengWu for 8 forecast steps starting from the ERA5 reanalysis field [42] at 00:00 UTC on December 30th, 2021.

To construct the flow-dependent ensemble, we follow a time-lagging strategy [36, 35]. We use ERA5 reanalysis fields at 6-hour intervals from 06:00 UTC on December 30th to 18:00 UTC on December 31st, 2021. Each of these 7 states is forecasted forward using FengWu for 7 to 1 steps respectively, so that all ensemble members align temporally at 00:00 UTC on January 1st. Combined with the original background forecast, we obtain a total of 8 ensemble members. The deviations of these members from their ensemble mean form the flow-dependent perturbation samples used to finetune the LoRA modules in the decoder of the pretrained VAE.

After finetuning, we perform variational data assimilation using simulated observations sampled from ERA5 analysis fields at 1000 random points. To assess the contribution of LoRA-based finetuning, we compare two assimilation results: (1) one using the frozen VAE decoder (without LoRA finetuning), (2) the other using the adaptively finetuned decoder (LoRA-EnVar). We compute the analysis error for both methods and visualize the spatial error difference between the two, i.e., $\mathrm{Error}_{\mathrm{no-tune}} - \mathrm{Error}_{\mathrm{LoRA-EnVar}}$. Positive values indicates improvement due to LoRA adaptation.

Figure 2 presents the results for three key variables: mean sea level pressure (mslp), geopotential at $500\,\mathrm{hPa}$ (z500), and geopotential at $850\,\mathrm{hPa}$ (z850). The most significant improvement is observed over the northeastern Atlantic west of the UK, a region known for strong jet streams and frequent mid-latitude cyclones during winter. These dynamically active conditions lead to highly flow-dependent forecast errors. The localized improvement suggests that LoRA-based adaptation helps capture such transient error structures more effectively than static background models, thereby enhancing assimilation accuracy in regions of rapid atmospheric variability.

## 4.3 Cyclic Assimilation and Forecasting

To evaluate the long-term performance of LoRA-EnVar in a near-realistic operational setting, we conduct a cyclic assimilation and forecasting experiment over a full month, starting from 00:00 UTC on January 1st, 2022. Following [19, 43], the system performs one assimilation cycle and produces an updated analysis every 6 hours, which is used to initialize the next forecast.

**Background Ensemble Generation**  We follow the framework illustrated in Figure 1, using an ensemble size of 8 (see the Appendix for experiments with a different ensemble size). Unlike the single-step experiment where ensemble members were generated using ERA5 reanalysis fields, here we construct the ensemble entirely from within the system, without access to external reanalysis data. Specifically, we use a time-lagging strategy: past analysis fields produced by the system itself are used to initialize forecasts that form the ensemble (see the Appendix for details). This design ensures full self-consistency within the cyclic forecast-assimilation loop.

**Observation Settings**  The cyclic assimilation experiment uses the same 1000 fixed synthetic observation stations as in the single-step setup, with full-variable measurements derived from ERA5. We have also conducted experiments on GDAS real-world observations [44], with results reported in the Appendix.

**Compute Resources**  Experiments are performed on a single NVIDIA A100 GPU, with each assimilation cycle requiring approximately 14 seconds for finetuning and 10 seconds for assimilation.

**LoRA vs. Traditional Algorithms**  We compare LoRA-EnVar against three baselines: (1) 3DVar, which uses a static climatological background covariance following [17]; (2) Hybrid 3DEnVar, which combines static and flow-dependent covariances with an optimized coefficient $\alpha$, following [45]; (3) VAE-Var (no finetune) [19], which uses the pretrained VAE decoder without any LoRA finetuning.

Figure 3 reports the RMSE and bias of key variables over the 31-day period. The results show that: LoRA-EnVar outperforms VAE-Var (no finetune) across all variables and lead times, confirming that flow-dependent finetuning of the generative model improves assimilation accuracy; LoRA-EnVar achieves lower RMSE and bias than Hybrid 3DEnVar, particularly in geopotential and wind fields, demonstrating its advantage over conventional hybrid schemes; 3DVar performs worst, as expected, due to its inability to adapt to real-time flow conditions. These findings indicate that LoRA-EnVar not only effectively integrates climatological and flow-dependent information in a non-Gaussian setting but also remains stable and accurate in long-term forecasting scenarios.

**LoRA vs. Full Finetuning**  To assess the effectiveness of LoRA-based online adaptation, we compare it against two variants of full finetuning strategies applied to the VAE decoder: (1) Full finetune (w/o reset): At each assimilation step, the decoder parameters are inherited from the previous cycle and updated using flow-dependent ensemble samples; (2) Full finetune (w/ reset): At each step, the decoder is reloaded from the original pretrained model before being finetuned using current ensemble samples. These two strategies represent two ends of the full-parameter update spectrum: the former prioritizes continuity but risks overfitting or catastrophic forgetting, while the latter improves stability at the expense of temporal coherence and increased computational cost.

Figure 4 presents a comparative evaluation of the three finetuning strategies across multiple key variables. It can be seen that LoRA-EnVar maintains consistently low RMSE and bias over time, while both full finetuning approaches exhibit notable trade-offs. The "Full finetune (w/o reset)" strategy, which accumulates parameter updates across cycles, tends to degrade in performance for certain variables, most prominently Z500 and t2m, likely due to catastrophic forgetting of the climatological structure learned during pretraining. In contrast, the "Full finetune (w/ reset)" strategy avoids such degradation by reloading the pretrained decoder at each step, but this comes at the cost of a higher computational burden, as it involves updating the full decoder parameters repeatedly.

Despite its simplicity, LoRA-EnVar achieves comparable or even superior performance to both full finetuning variants. This demonstrates that low-rank adaptation offers a highly efficient way to incorporate flow-dependent corrections while preserving the stability and generalization capabilities inherent to the pretrained model.

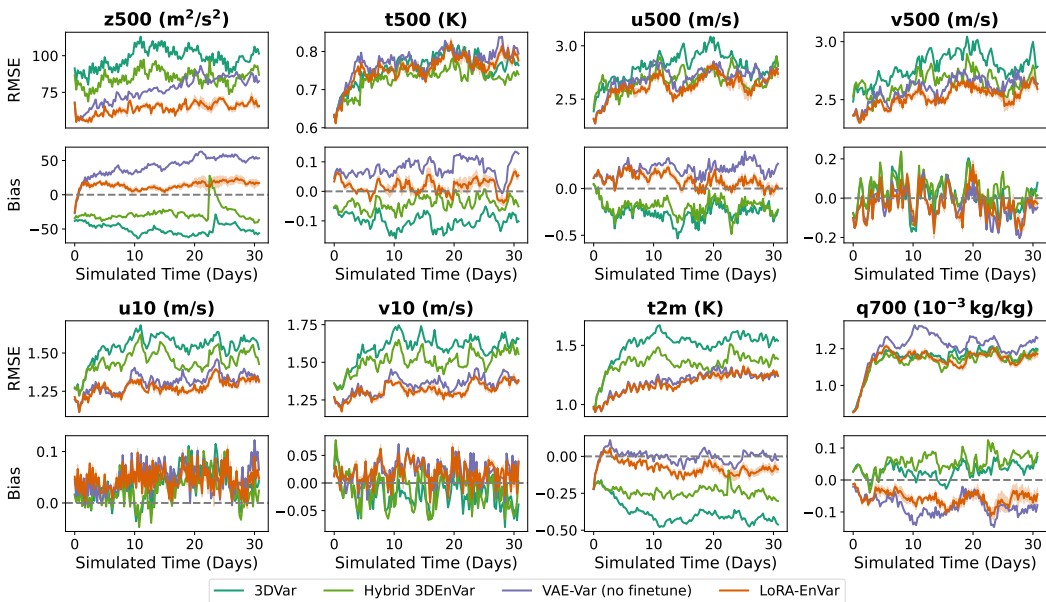

Figure 3: **RMSE and bias of key atmospheric variables over a 30-day cyclic assimilation experiment.** Results are shown for 3DVar, Hybrid 3DEnVar, VAE-Var (no finetune), and LoRA-EnVar, respectively. LoRA-EnVar experiments are repeated five times with different random seeds, and we report the mean along with the 1-$\sigma$ standard deviation.

**Different Observation Densities** Motivated by the results under 1000 observation points, we further evaluate the performance of LoRA-EnVar under varying levels of observation availability by conducting cyclic assimilation experiments with 250, 500, 1000, and 2000 observations per cycle, corresponding to approximately 0.024%, 0.048%, 0.096%, and 0.19% of the total grid points, respectively. In all cases, observation locations are fixed and provide values for all variables.

Figure 5 shows the RMSE of z500 and u850 under these different settings. Across all observation densities, LoRA-EnVar consistently outperforms the non-finetuned VAE-Var model, indicating that flow-dependent adaptation provides benefits regardless of how sparse or dense the observational data are. As expected, the relative performance gain from LoRA-EnVar tends to decrease as the observation density increases. When more observations are assimilated, the analysis becomes increasingly constrained by observational data, reducing the marginal impact of more accurate background error modeling. Nonetheless, even in the most observation-rich case (2000 points), LoRA-EnVar continues to show improvements over both VAE-Var and Hybrid 3DEnVar, demonstrating its strong adaptability and effectiveness in a variety of observational scenarios.

### 4.4 Evaluation of Parameter Efficiency

In our implementation, full finetuning requires updating all decoder parameters at every assimilation cycle, totaling 431,772,613 parameters. In contrast, LoRA-EnVar only introduces low-rank adapters into the query projection layers of the Swin Transformer blocks. With a fixed LoRA rank of 2, the total number of tunable parameters in our setup is just 165,888. This represents a reduction of over 99.96% in the number of updated parameters, while still achieving comparable or superior assimilation performance as shown in Section 4.3. This highlights the practical advantage of LoRA-EnVar for scalable, long-term deployment in operational settings, where both model size and computational efficiency are critical concerns.

## 5 Related Work

**AI for Data Assimilation** Beyond the approaches mentioned in the introduction, numerous studies have applied AI techniques to improve DA, including latent-space assimilation [46, 47, 48, 49, 50,

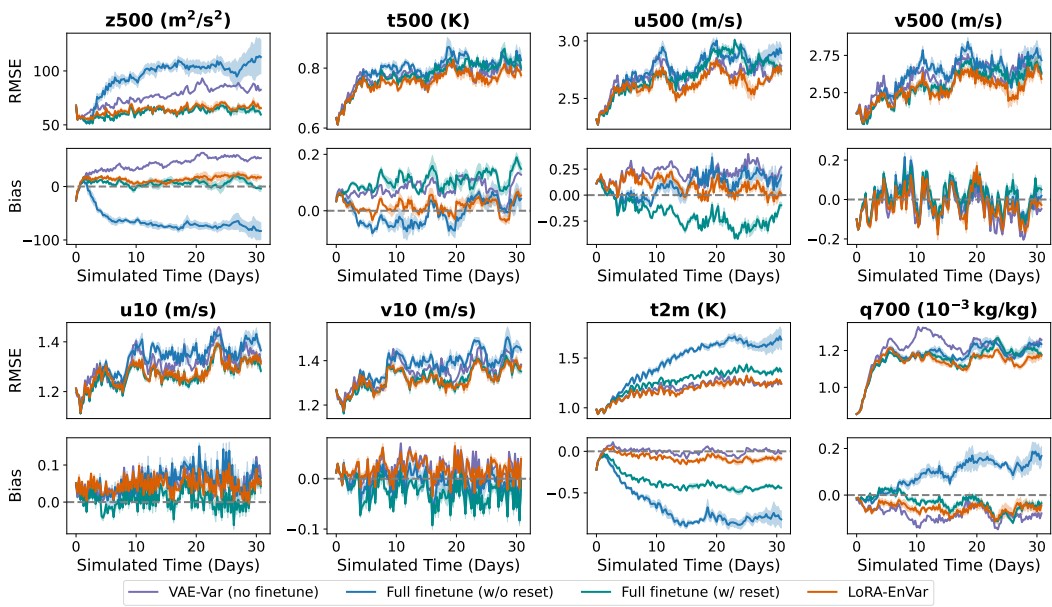

Figure 4: **Comparison of finetuning strategies in cyclic assimilation.** RMSE and bias are reported for LoRA-EnVar, VAE-Var (no-finetune) and two full-finetune baselines under the same experimental setting. LoRA-EnVar and two full-finetune experiments are repeated five times with different random seeds, and we report the mean along with the 1-$\sigma$ standard deviation.

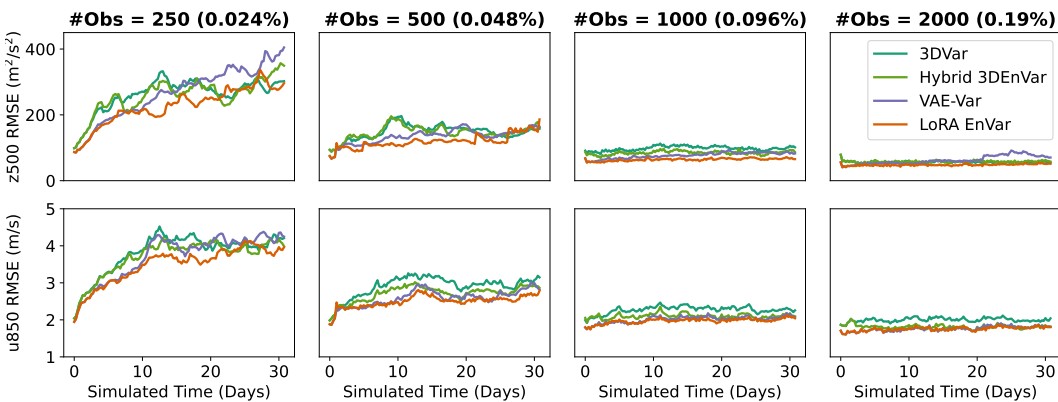

Figure 5: **RMSE of z500 and u850 under varying observation densities in cyclic assimilation.** Results are shown for four observation counts, comparing four different DA methods.

51], directly learning the assimilation mapping [52, 53], neural process [54], and reinforcement learning [55]. These methods aim to enhance assimilation accuracy or computational efficiency. However, most are limited to small-scale problems and often omit the flow-dependent nature of background errors, making them less applicable to high-resolution, operational forecasting scenarios.

**LoRA for Dynamical Systems**  Recent studies have explored the use of LoRA in learning dynamical systems, primarily in the context of transfer learning for PDE solvers. These approaches leverage structural similarities between systems to enable parameter-efficient adaptation, and are often evaluated on simulated or idealized settings [56, 57]. In contrast, our work applies LoRA to a real-world atmospheric system, focusing not on solving the governing equations directly, but on modeling the evolution of forecast uncertainty within a DA framework. To the best of our knowledge, this

is the first use of LoRA for learning flow-dependent background error dynamics in operational-scale weather forecasting.

## 6 Conclusion

We proposed LoRA-EnVar, a hybrid ensemble variational DA framework that combines deep generative modeling with parameter-efficient flow-dependent adaptation. Our method captures both climatological and real-time error structures, achieving improved assimilation accuracy while reducing the number of trainable parameters by three orders of magnitude. By enabling online adaptation of background error distributions at minimal computational cost, LoRA-EnVar offers a scalable solution for next-generation data assimilation systems. It is particularly well suited for high-resolution numerical weather prediction settings, where both modeling accuracy and efficiency are critical. This work contributes toward bridging machine learning advances with operational DA workflows, and opens new directions for learning-based non-Gaussian assimilation methods in atmospheric sciences.

Despite its strong empirical performance, LoRA-EnVar currently depends on heuristically chosen finetuning parameters, and lacks a theoretical characterization of its adaptation dynamics. Moreover, it inherits the intrinsic limitation of variational assimilation in producing only point estimates, leaving uncertainty propagation an open challenge. Future work may address these issues by studying its theoretical properties and extending it toward fully probabilistic assimilation frameworks.

## Acknowledgments

This work is supported by the National Natural Science Foundation of China (NO.U2242210). This work is also supported by the Shanghai Science and Technology Commission Project and Shanghai Artificial Intelligence Laboratory.

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

# A  Variational Objective for LoRA-EnVar

Once the VAE is trained or finetuned within the LoRA-EnVar framework, we employ its decoder for variational data assimilation. Similar to traditional variational assimilation, we optimize latent variables $\mathbf{z}$ to obtain the posterior estimate. However, in contrast to linear Gaussian models, the mapping from latent space to physical state is non-linear, given by:

$$\mathbf{x} = \mathcal{D}(\mathbf{z}) + \mathbf{x}_b, \tag{4}$$

where $\mathcal{D}(\mathbf{z})$ denotes the output of the trained decoder, and $\mathbf{x}_b$ is the background state.

The observation term is accordingly reformulated as:

$$\tilde{\mathcal{L}}_o(\mathbf{z}) = \mathcal{L}_o\left(\mathcal{D}(\mathbf{z}) + \mathbf{x}_b, \mathbf{y}\right), \tag{5}$$

where $\mathcal{L}_o$ denotes the original observation mismatch loss, typically based on squared error under Gaussian observation assumptions; in our work, it is formulated as follows:

$$\mathcal{L}_o(\mathbf{x}, \mathbf{y}) = \frac{1}{2}(\mathbf{y} - \mathcal{H}(\mathbf{x}))^{\mathrm{T}}\mathbf{R}^{-1}(\mathbf{y} - \mathcal{H}(\mathbf{x})), \tag{6}$$

where $\mathcal{H}$ is an observation operator, mapping variables from the physical space to the observation space.

From a theoretical perspective, a nonlinear transformation from latent to physical space introduces a non-constant Jacobian determinant in the background term. Since this term is intractable in practice, we approximate the background regularization as:

$$\tilde{\mathcal{L}}_b(\mathbf{z}) = \frac{1}{2}\lambda\mathbf{z}\mathbf{z}^{\mathrm{T}} \tag{7}$$

where $\lambda$ is a positive scalar used to adjust the strength of the prior constraint. In our work, the latent dimension is half of the dimension of the physical space; therefore, $\lambda$ is empirically set to 4.

The total variational loss is thus:

$$\tilde{\mathcal{L}}(\mathbf{z}) = \frac{1}{2}\lambda\mathbf{z}\mathbf{z}^{\mathrm{T}} + \frac{1}{2}(\mathbf{y} - \mathcal{H}\left(\mathcal{D}(\mathbf{z}) + \mathbf{x}_b\right)^{\mathrm{T}}\mathbf{R}^{-1}(\mathbf{y} - \mathcal{H}\left(\mathcal{D}(\mathbf{z}) + \mathbf{x}_b\right)). \tag{8}$$

All components of this objective are differentiable, allowing efficient gradient-based optimization. During assimilation, we fix the decoder parameters and perform latent-space optimization using the L-BFGS algorithm. The full procedure is summarized in Algorithm 1. This latent-space assimilation

---

**Algorithm 1** LoRA-EnVar Assimilation Step

---

**Require:** Trained and finetuned decoder $\mathcal{D}_i$ at time $T_i$, background state $\mathbf{x}_b$, observation $\mathbf{y}$

Initialize latent vector: $\mathbf{z} \leftarrow \mathbf{0}$

Compute background term: $\tilde{\mathcal{L}}_b(\mathbf{z}) \leftarrow \frac{1}{2}\lambda\mathbf{z}^{\mathrm{T}}\mathbf{z}$

Compute observation term: $\tilde{\mathcal{L}}_o(\mathbf{z}) \leftarrow \mathcal{L}_o(\mathcal{D}_i(\mathbf{z}) + \mathbf{x}_b, \mathbf{y})$

Compute total loss term: $\tilde{\mathcal{L}}(\mathbf{z}) = \tilde{\mathcal{L}}_b(\mathbf{z}) + \tilde{\mathcal{L}}_o(\mathbf{z})$

Obtain the derivative $\frac{\mathrm{d}\tilde{\mathcal{L}}(\mathbf{z})}{\mathrm{d}\mathbf{z}}$ using auto-differentiation [58] and minimize total loss via L-BFGS [59]

Return analysis: $\mathbf{x}_a = \mathcal{D}_i(\mathbf{z}^\star) + \mathbf{x}_b$

---

framework allows us to leverage the expressive power of deep generative models while maintaining tractable optimization.

# B  Implementation of Hybrid 3DEnVar

In our experiments, we implement a hybrid 3DEnVar system by incorporating both static climatological and ensemble-derived flow-dependent covariances into the variational assimilation framework. Our implementation is based on the control variable extension method proposed in [60, 14], and theoretically justified in [61], which proves its equivalence to earlier formulations based on covariance matrix blending [9].

In hybrid 3DEnVar, the covariance matrix $\mathbf{B}$ is expressed as a linear combination:

$$\mathbf{B} = \alpha\mathbf{B}_s + (1-\alpha)\mathbf{B}_e, \tag{9}$$

where $\mathbf{B}_s$ is a static, climatological covariance (usually estimated offline), and $\mathbf{B}_e$ is derived from flow-dependent ensemble perturbations.

To avoid directly computing and inverting the blended $\mathbf{B}$, we follow the extended control variable formulation. The analysis increment $\delta\mathbf{x}$ is expressed as a weighted sum:

$$\delta\mathbf{x} = \beta_1\mathbf{B}_s^{1/2}\mathbf{v}_1 + \beta_2\mathbf{B}_e^{1/2}\mathbf{v}_2, \tag{10}$$

where $\mathbf{v}_1$ and $\mathbf{v}_2$ are control variables defined in the transformed space, and $\mathbf{B}_s^{1/2}$ and $\mathbf{B}_e^{1/2}$ are square-root operators of the respective covariance matrices. The corresponding cost function becomes:

$$\mathcal{L}_{\text{hybrid}-3\text{DEnVar}}(\mathbf{v}_1, \mathbf{v}_2) = \frac{1}{2}\mathbf{v}_1^{\mathrm{T}}\mathbf{v}_1 + \frac{1}{2}\mathbf{v}_2^{\mathrm{T}}\mathbf{v}_2 + \frac{1}{2}\left[\mathbf{y} - \mathcal{H}\left(\mathbf{x}_b + \delta\mathbf{x}\right)\right]^{\mathrm{T}}\mathbf{R}^{-1}\left[\mathbf{y} - \mathcal{H}\left(\mathbf{x}_b + \delta\mathbf{x}\right)\right]. \tag{11}$$

This formulation allows preconditioning of each component and efficient optimization, while maintaining the desired covariance blending effect in the physical space. It has been shown that the control variable formulation is theoretically equivalent to the covariance blending formulation when the weights satisfy $\beta_1 = \sqrt{\alpha}$ and $\beta_2 = \sqrt{1-\alpha}$ [45]. In our experiments, we find that the best performance is achieved when $\beta_2 = 0.6$, corresponding to $\alpha = 0.64$.

In our implementation of Hybrid 3DEnVar, both $\mathbf{B}_s$ and $\mathbf{B}_e$ are constructed using the GEN_BE tool from the WRF Data Assimilation system (WRFDA) [17, 19]. The control variables $\mathbf{v}_1$ and $\mathbf{v}_2$ are optimized jointly using a quasi-Newton solver (L-BFGS). The final analysis increment is reconstructed by combining the two transformed increments, and the analysis state is updated as:

$$\mathbf{x}_a = \mathbf{x}_b + \delta\mathbf{x}. \tag{12}$$

## C   Experimental Details

### C.1   Neural Network Architectures

Our VAE architecture is inspired by the design in VAE-Var [19], with modifications tailored to realistic high-resolution atmospheric modeling. Specifically, we adopt a backbone structure similar to that of the FengWu forecasting model. As illustrated in Figure 6, both the encoder and decoder are built upon the Swin Transformer V2 [39] architecture.

Each of the encoder and decoder components consists of three stacked modules: an input encoder, a central transformer, and an output decoder. All three modules contain attention layers. To enable parameter-efficient adaptation, we insert LoRA modules into the query projections of all attention layers within these components.

Although only the decoder is used during assimilation, both the encoder and decoder are equipped with LoRA modules during training, and their parameters are jointly optimized to ensure architectural symmetry and effective latent encoding. During the assimilation phase, the encoder is not used and the decoder, including both its backbone and LoRA components, is kept fixed and used to transform optimized latent variables into physical states.

### C.2   Dataset Construction

**Historical Error Samples Generation**   During the offline training phase, we adopt the classical NMC method (originating from the National Meteorological Center) to generate historical forecast error samples, as outlined in Algorithm 2. Reanalysis data are treated as the reference ("true") states. For each training sample, we select two reanalysis states separated by a time gap $\tau$ and forecast both forward to the same target time—once using a single-step forecast from $\mathbf{x}_\tau$, and once using a two-step forecast starting from $\mathbf{x}_0$. The difference between these two forecasts approximates the background forecast error. Repeating this procedure across a long historical time window allows us to construct a representative training set. In our implementation, we use ERA5 reanalysis data from 1979 to 2015.

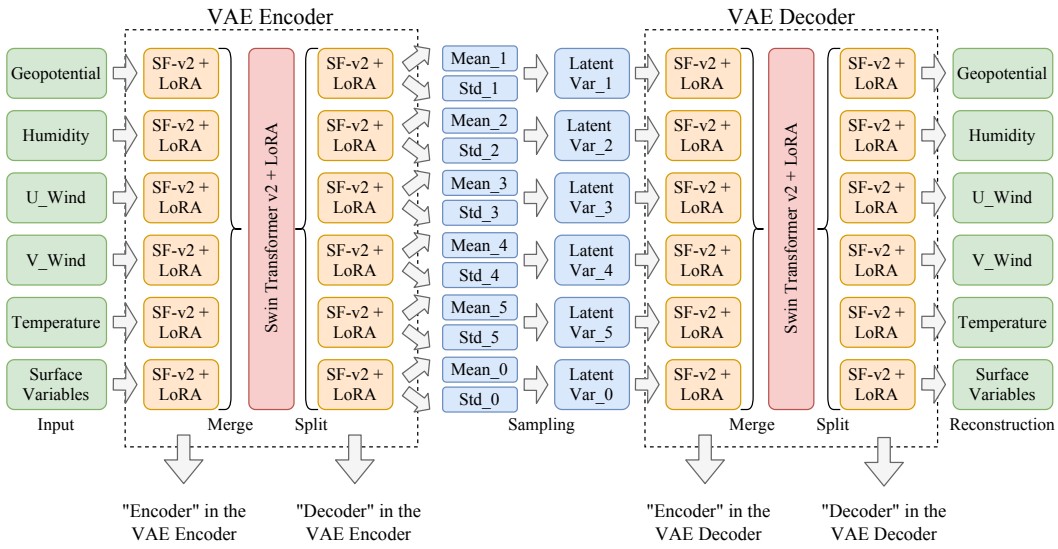

Figure 6: **Overview of the VAE architecture used in LoRA-EnVar.** Both the encoder and decoder are based on Swin Transformer V2 blocks, and LoRA modules are inserted in all query projections.

---

**Algorithm 2** NMC-Based Training Set Construction

---

**Require:** Forecast model $\mathcal{M}$, number of samples $N$, forecast interval $\tau$

    **for** $i = 1$ to $N$ **do**

        Select two reanalysis states: $\mathbf{x}_0$, $\mathbf{x}_\tau$ (separated by $\tau$)

        $\hat{\mathbf{x}}_1 \leftarrow \mathcal{M}_{\tau \to 2\tau}(\mathbf{x}_\tau)$

        $\hat{\mathbf{x}}_2 \leftarrow \mathcal{M}_{\tau \to 2\tau} \circ \mathcal{M}_{0 \to \tau}(\mathbf{x}_0)$

        Append forecast difference $\hat{\mathbf{x}}_1 - \hat{\mathbf{x}}_2$ to training set

    **end for**

---

**Flow-Dependent Error Samples Generation**    To generate flow-dependent error samples in a self-consistent manner during cycling assimilation, we adopt a time-lagging ensemble strategy, illustrated in Figure 7. For each assimilation cycle, ensemble members are constructed by forecasting from previous analysis times, with varying lead times (e.g., 1, 2, or 3 steps), such that all members arrive at the same target time. This approach enables ensemble generation without reliance on external forecasts or reanalysis data, and allows the system to remain close-loop during long-term assimilation.

However, in practice, forecast errors exhibit magnitude dependence on lead time, that is, members with longer lead times tend to have larger absolute deviations. Directly mixing such samples can cause the model to prioritize high-magnitude signals and distort the representation of flow-dependent structure.

To address this, we apply a two-stage normalization process:

- **Inter-member normalization**: For each ensemble member, we compute the standard deviation of every variable at each vertical level. The corresponding error sample is then divided by this standard deviation, ensuring that all members have comparable amplitude across variables, while preserving their structural differences.

- **Standard Gaussian normalization**: After aligning magnitudes across members, we apply global standardization (zero mean, unit variance) to the full batch of normalized error samples before using them to finetune the decoder via LoRA.

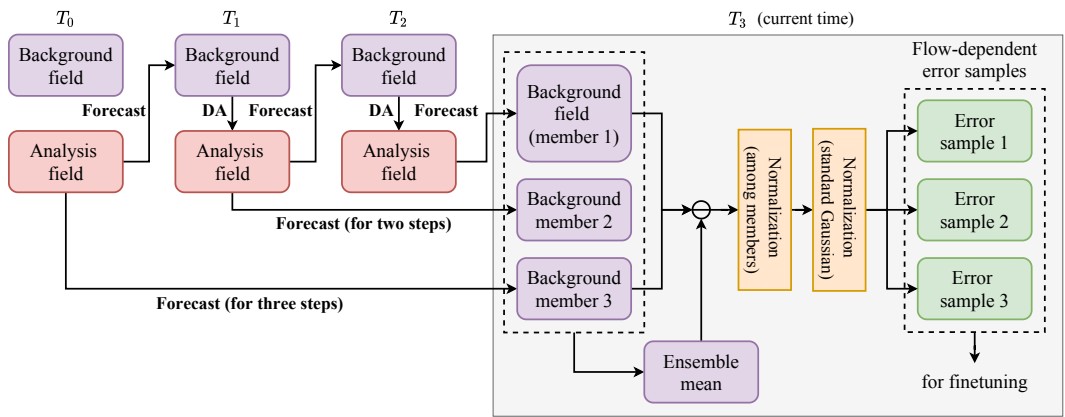

Figure 7: **Illustration of time-lagging ensemble construction with normalization.** Forecasts with different lead times are aligned to the same target time. Member-specific normalization is applied before finetuning to mitigate magnitude imbalance across lead times.

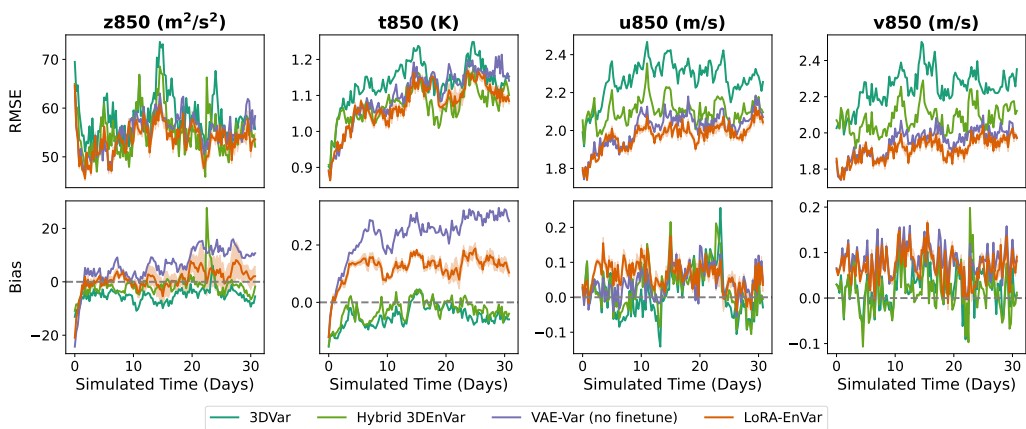

Figure 8: **RMSE and bias comparisons at 850 hPa.** LoRA-EnVar and three baseline methods (3DVar, hybrid 3DEnVar, and VAE-Var without finetuning) are compared. LoRA-EnVar shows a consistent advantage in both accuracy and bias reduction.

### C.3 Training & Finetuning Hyperparameters

The VAE is trained using a composite loss function of the form:

$$Loss = \frac{1}{\sigma^2}\mathcal{L}_{rec} + \mathcal{L}_{KL}, \tag{13}$$

where $\mathcal{L}_{rec}$ is the reconstruction loss, $\mathcal{L}_{KL}$ is the Kullback–Leibler divergence between the latent posterior and prior, and $\sigma$ is a positive scaling factor. In our experiments, we set $\sigma = 2.0$.

Both training and finetuning procedures use the Adam optimizer [62]. The learning rate for offline VAE training is set to $10^{-4}$. During finetuning, we use a learning rate of $10^{-5}$ for full model finetuning, and $10^{-2}$ for LoRA-based finetuning. In both cases, finetuning is performed for five epochs.

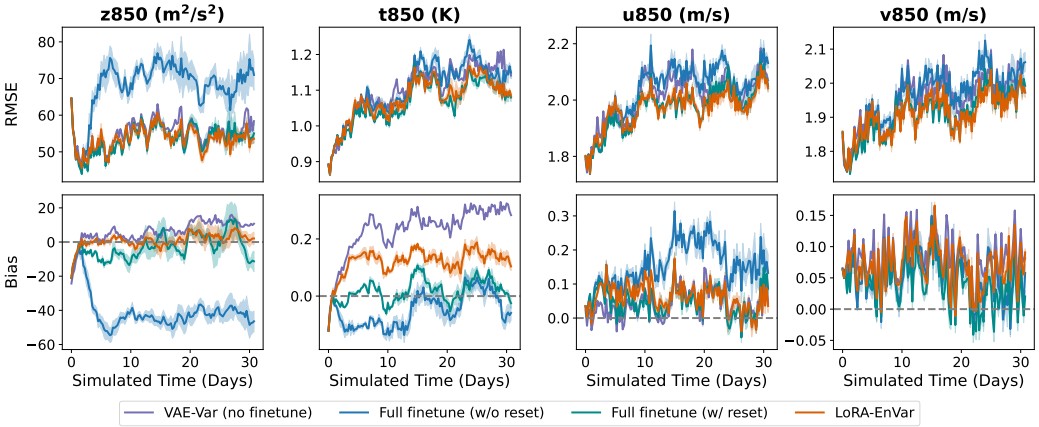

Figure 9: **RMSE and bias comparisons at 850 hPa.** Different finetuning strategies (LoRA-EnVar, full finetuning without resetting, and full finetuning with per-cycle reset) are compared. LoRA-EnVar consistently achieves strong accuracy with reduced bias drift.

# D  Additional Results

## D.1  Supplementary Results for Main Experiments

**Cyclic Assimilation and Forecasting**    To further validate the effectiveness of LoRA-EnVar, we provide a supplementary assessment on additional atmospheric levels. Figure 8 reports the RMSE and bias for four variables at 850 hPa - geopotential (z850), temperature (t850), zonal wind (u850) and meridional wind (v850) - in a 30-day cyclic assimilation experiment.

We compare LoRA-EnVar with three baselines: classical 3DVar, hybrid 3DEnVar, and VAE-Var without finetuning. Consistent with the main results at other levels, LoRA-EnVar achieves the lowest RMSE and bias across most variables throughout the simulation period. Notably, the improvement is most pronounced for wind components (u850 and v850), highlighting the model's ability to capture flow-dependent structures that are particularly dynamic in the lower troposphere.

To further isolate the effect of different finetuning strategies, we compare LoRA-based adaptation with two variants of full-model finetuning: one that accumulates updates over time (w/o reset) and another that resets the decoder weights at each cycle (w/ reset). As shown in Figure 9, LoRA-EnVar achieves comparable or better performance than both full finetuning approaches, with better stability over time and lower bias accumulation.

**Different Observation Densities**    To complement the main paper's analysis, we report additional assimilation results across different observation densities (250, 500, 1000, and 2000 observations, corresponding to approximately 0.024%, 0.048%, 0.096%, and 0.19% of total grid points). These results extend the evaluation in Section 4.3 by including more key atmospheric variables.

Figures 10–12 show RMSE curves over a 30-day assimilation period at various vertical levels (surface, 500 hPa, and 850 hPa), comparing four methods: 3DVar, Hybrid 3DEnVar, VAE-Var without finetuning, and LoRA-EnVar. The comparison includes surface-level variables (t2m, u10, v10, mslp), as well as free-atmosphere variables at 500 hPa and 850 hPa (z, u, v, t, q).

While performance varies across different variables and observation settings, LoRA-EnVar generally outperforms the baselines in most cases, particularly under sparse observation scenarios. In denser cases, it remains competitive with traditional methods while offering better parameter efficiency.

## D.2  Impact of Ensemble Generation Strategy

In this section, we investigate the impact of background ensemble generation strategies on the performance of LoRA-EnVar. While the main paper uses a time-lagging ensemble with 8 members,

here we explore an extended 20-member ensemble that combines two distinct sources of flow-dependent perturbations.

The first part of the ensemble includes 12 members generated via the standard time-lagging method, an expansion over the 8 members used in the main setting. The second part consists of 8 additional members derived from historical forecasts initialized on the same calendar dates one year earlier. Specifically, to construct the ensemble for, e.g., January 3, 2022, we collect forecasts targeting this date from initialization times between January 1 and January 2, 2021, using varying lead times (1 to 8 steps). This design leverages the seasonal similarity of atmospheric conditions across years to enhance ensemble diversity.

Figures 13 and 14 show the RMSE and bias of LoRA-EnVar and baseline methods under this expanded ensemble setting. Overall, LoRA-EnVar continues to outperform traditional assimilation methods and full finetuning variants on most variables. The increase in ensemble size also leads to a general reduction in assimilation errors for all ensemble-based methods, highlighting the benefit of richer ensemble sampling.

### D.3 Impact of LoRA Rank

To evaluate the sensitivity of LoRA-EnVar to the rank hyperparameter in LoRA modules, we conduct a series of cyclic assimilation experiments with different LoRA ranks: 1, 2 (default in the main paper), 4, 8, and 16. All other configurations, including model architecture, observation density, and ensemble setup, are kept identical to the main setting.

As shown in Figure 15, when the LoRA rank is set to 1, the assimilation accuracy slightly decreases, especially in terms of bias for certain variables such as near-surface temperature and u500. This suggests that a rank of 1 may not provide sufficient representational capacity to capture flow-dependent forecast corrections. In contrast, when the rank is set to 2 or higher, the assimilation results remain largely consistent, with negligible differences in both RMSE and bias across ranks 2, 4, 8, and 16.

These findings indicate that a LoRA rank of 2 provides adequate modeling capacity for our use case, while larger ranks offer no significant gain but incur additional computational overhead. This highlights the robustness and parameter efficiency of LoRA-EnVar.

### D.4 Assimilation with Real Observations

To further evaluate the practical effectiveness of LoRA-EnVar, we conduct additional experiments using real-world observations from the NOAA GDAS (Global Data Assimilation System) prepbufr dataset.

#### D.4.1 Experimental Setup

**Dataset Description**  The GDAS prepbufr dataset provides a comprehensive set of global meteorological observations curated by the National Centers for Environmental Prediction (NCEP). It contains surface and upper-air measurements collected from various sources, including land and marine stations, radiosondes, aircraft reports, and GTS (Global Telecommunications System) transmissions. The dataset also incorporates advanced remote sensing inputs such as wind profilers, radar-derived wind data from the U.S., satellite-derived winds from NESDIS, and ocean surface wind estimates from instruments like SSM/I. This rich and operationally relevant dataset serves as a benchmark in modern weather forecasting systems and is thus well-suited for evaluating real-data assimilation performance.

**Data Preprocessing**  Each entry in the prepbufr dataset represents an individual measurement recorded by a specific instrument at a known time and spatial location. Since our assimilation algorithm targets fixed assimilation times, we retain only observations that fall within a narrow time window, 30 minutes before and after each assimilation cycle. For example, at an assimilation time of 6 PM, we include only observations recorded between 5:30 PM and 6:30 PM. In addition, these observations are often irregularly located in space and do not lie directly on the forecast model's grid. To resolve this, we construct a new, finer observation grid, onto which the raw observations are projected using nearest-neighbor mapping. This intermediate grid shares the horizontal resolution of the FengWu model, but contains 40 vertical levels, much denser than the native FengWu grid,

to preserve the vertical structure while maintaining computational efficiency. After gridding, we perform quality control by discarding any observations that deviate significantly from the co-located ERA5 reanalysis values.

**Observation Operator Construction**    Once the observations are placed on the uniform grid, we define the observation operator $\mathcal{H}$ as the mapping from the FengWu forecast grid to this observation grid. This is implemented using a differentiable linear interpolation layer in PyTorch, ensuring full compatibility with our variational assimilation framework.

### D.4.2    Evaluation Protocols

To assess the effectiveness of LoRA-EnVar under realistic observational settings, we design two complementary evaluation protocols:

**ERA5-based Evaluation**    In this setting, we assimilate all available GDAS observations at each assimilation step and compare the resulting analysis fields with the ERA5 reanalysis. Specifically, we compute the RMSE between the analysis and ERA5 over the global domain, across multiple variables and levels. This evaluation reflects the system's ability to recover large-scale atmospheric structures and serves as a standard benchmark for assessing assimilation fidelity.

**Station-based Evaluation**    To more directly evaluate the predictive skill of the assimilation system at observation sites, we adopt a hold-out validation strategy. At each assimilation time, we randomly exclude 15% of the available observational stations from assimilation. The remaining 85% of observations are used to perform data assimilation as usual. After assimilation, we interpolate the resulting analysis fields to the locations of the held-out 15% stations and compute RMSE between the interpolated values and the actual observations at those sites. This protocol assesses the system's ability to generalize to unseen observations and is particularly relevant for evaluating performance in sparse or partially observed regions.

### D.4.3    Results and Analysis

**ERA5-based Evaluation Results**    In this context, we compare the analysis fields generated by different assimilation methods against the ERA5 reanalysis. To offer a clearer view of how assimilation improves upon raw observational input, we include a linear interpolation baseline that directly interpolates observations without consideration of model dynamics. As shown in Figure 16, all assimilation methods, including 3DVar, hybrid 3DEnVar, VAE-Var, and LoRA-EnVar, outperform linear interpolation across nearly all variables and time steps. This highlights the benefit of incorporating physical priors and model dynamics in reducing analysis error. Notably, among the model-based methods, LoRA-EnVar achieves consistently lower RMSE than VAE-Var and hybrid 3DEnVar, especially for variables like wind speed and temperature. These results align well with the trends observed in our earlier simulated observation experiments, reaffirming the effectiveness of LoRA-based flow-dependent adaptation in real-world settings.

**Station-based Evaluation Results**    To evaluate model generalization to unseen observations, we adopt a station hold-out strategy, where 15% of the GDAS observation stations are excluded from assimilation and used solely for evaluation. Figure 17 presents the RMSE of interpolated analysis values at these held-out stations. Due to the inherent noise and variability in real-world observations, these RMSE curves exhibit stronger fluctuations compared to ERA5-based evaluation. To highlight the performance difference between LoRA-EnVar and VAE-Var more clearly, we also plot the RMSE difference between the two methods (gray curves in Figure 17), where positive values indicate that LoRA-EnVar performs better at a given time step. Overall, we observe that LoRA-EnVar outperforms VAE-Var at most time steps and across most variables, indicating its superior ability to integrate sparse and noisy observational data without overfitting.

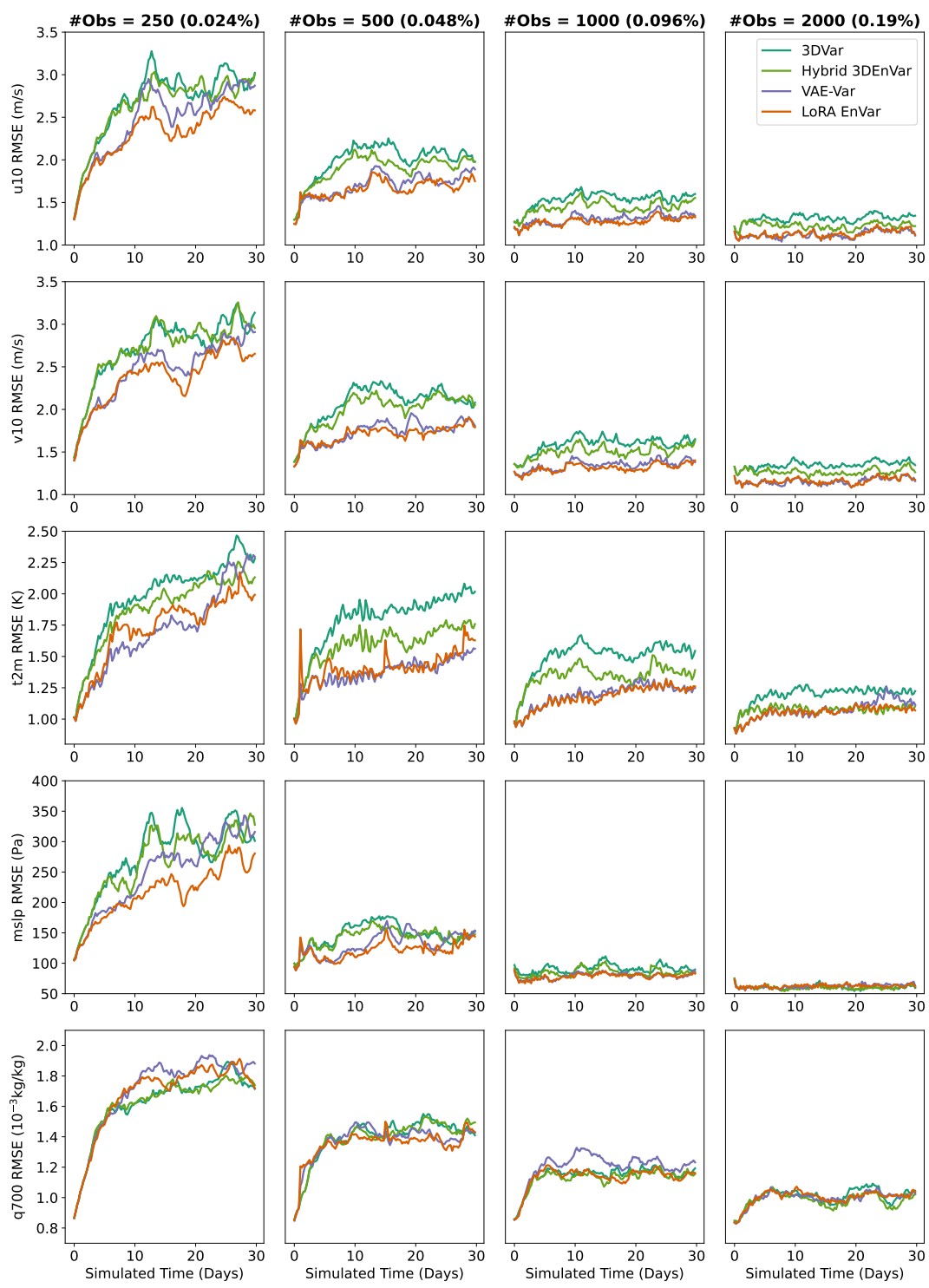

Figure 10: **RMSE comparisons of major surface-level variables and q700 under different observation densities (250∼2000 obs).** LoRA-EnVar is compared against 3DVar, Hybrid 3DEnVar, and VAE-Var (no finetuning).

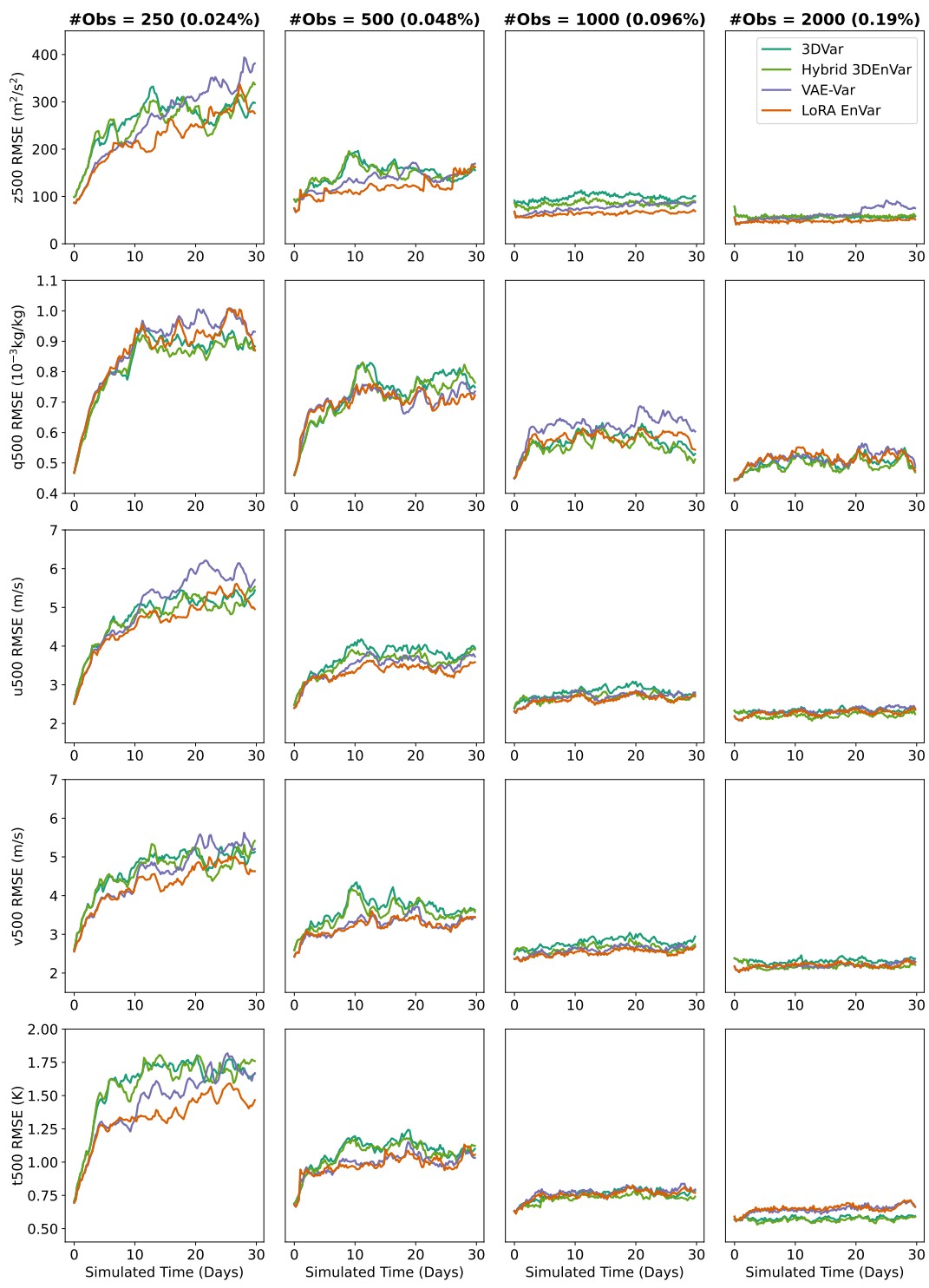

Figure 11: **RMSE comparisons at 500 hPa level for multiple variables under varying observation densities.** LoRA-EnVar is compared against 3DVar, Hybrid 3DEnVar, and VAE-Var (no finetuning).

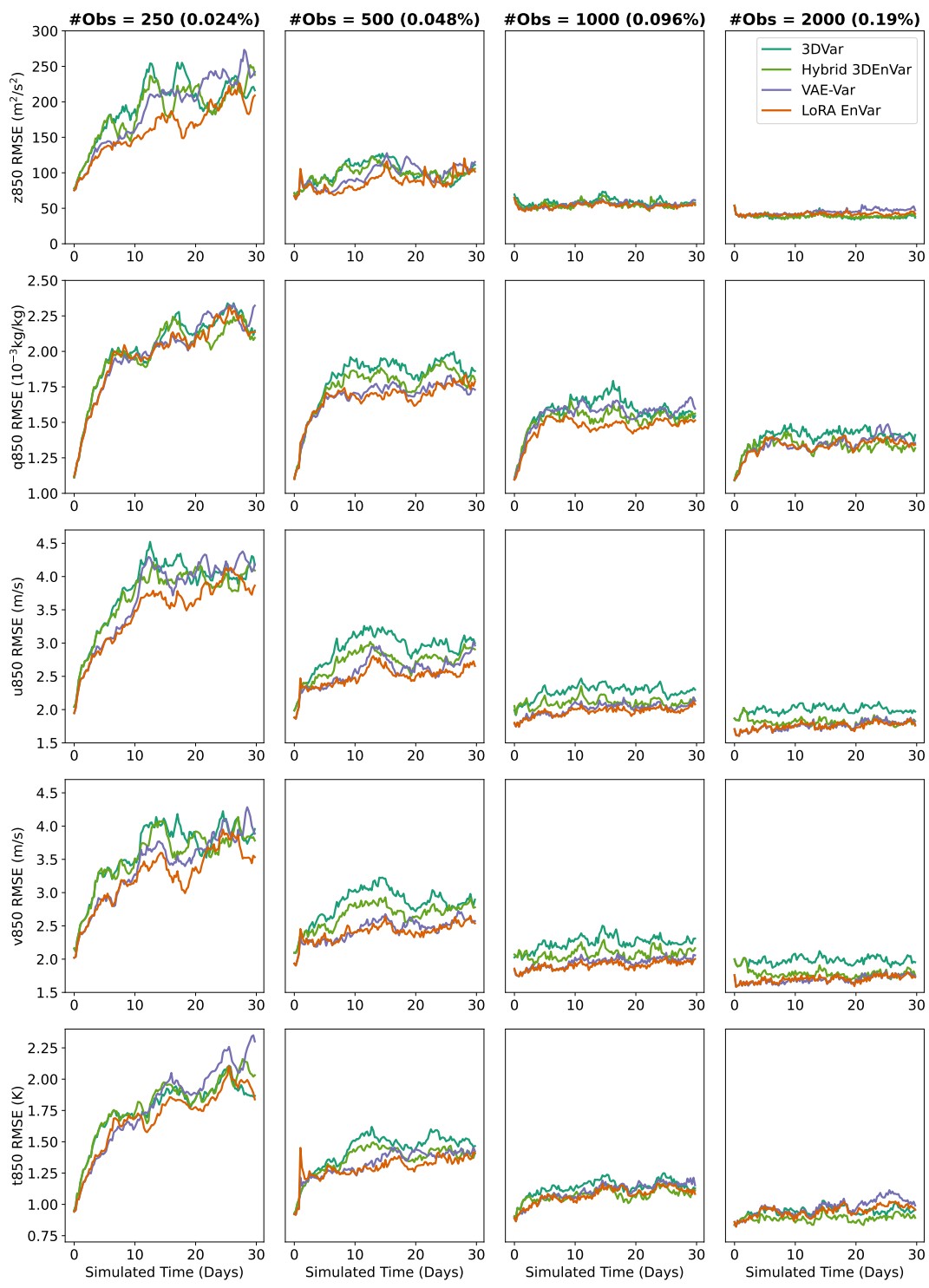

Figure 12: **RMSE comparisons at 850 hPa level for multiple variables under varying observation densities.** LoRA-EnVar is compared against 3DVar, Hybrid 3DEnVar, and VAE-Var (no finetuning).

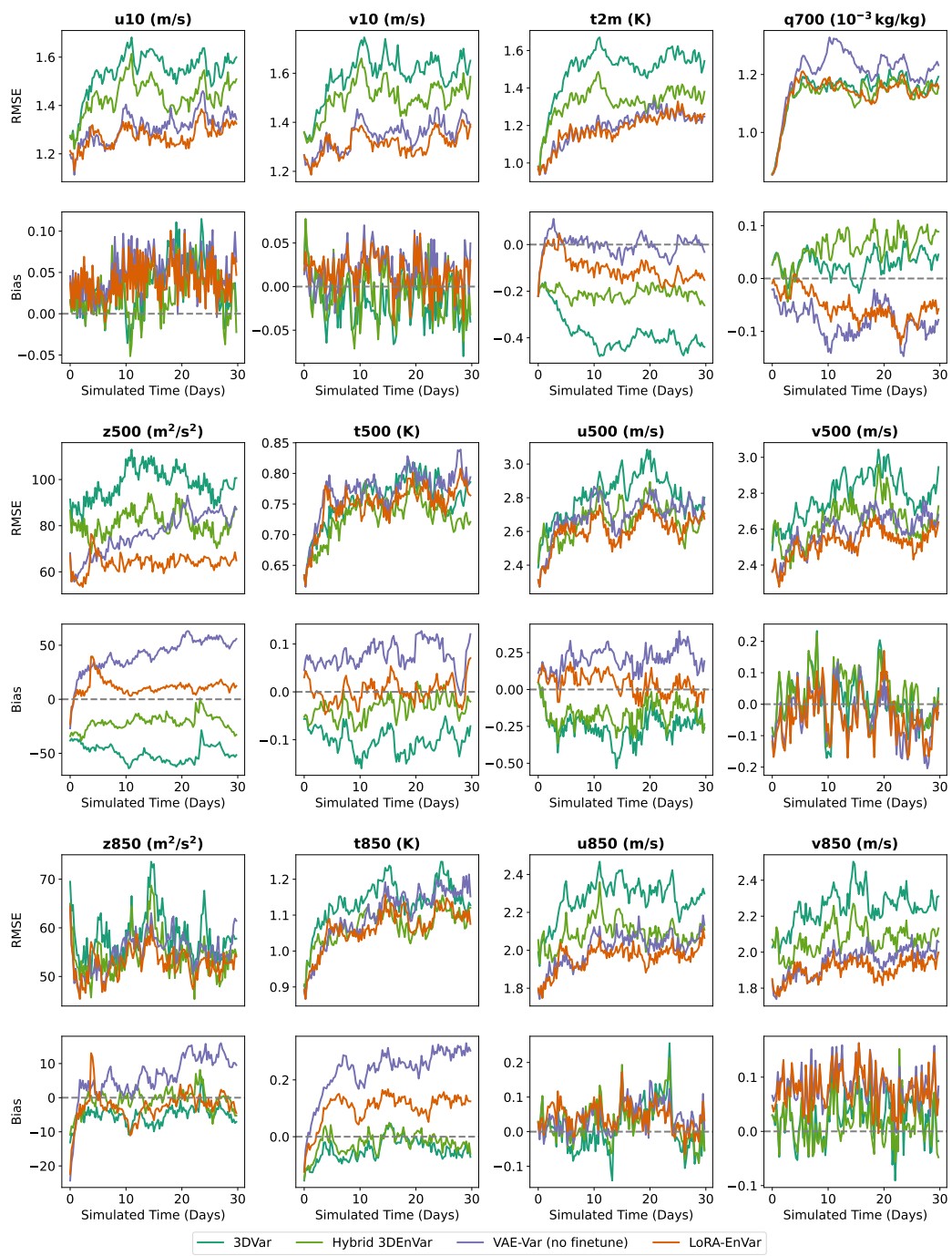

Figure 13: **RMSE and bias comparisons for major variables using a 20-member ensemble, including 12 time-lagged and 8 seasonally aligned historical forecasts.** LoRA-EnVar is compared with 3DVar, hybrid 3DEnVar, and VAE-Var (no finetuning).

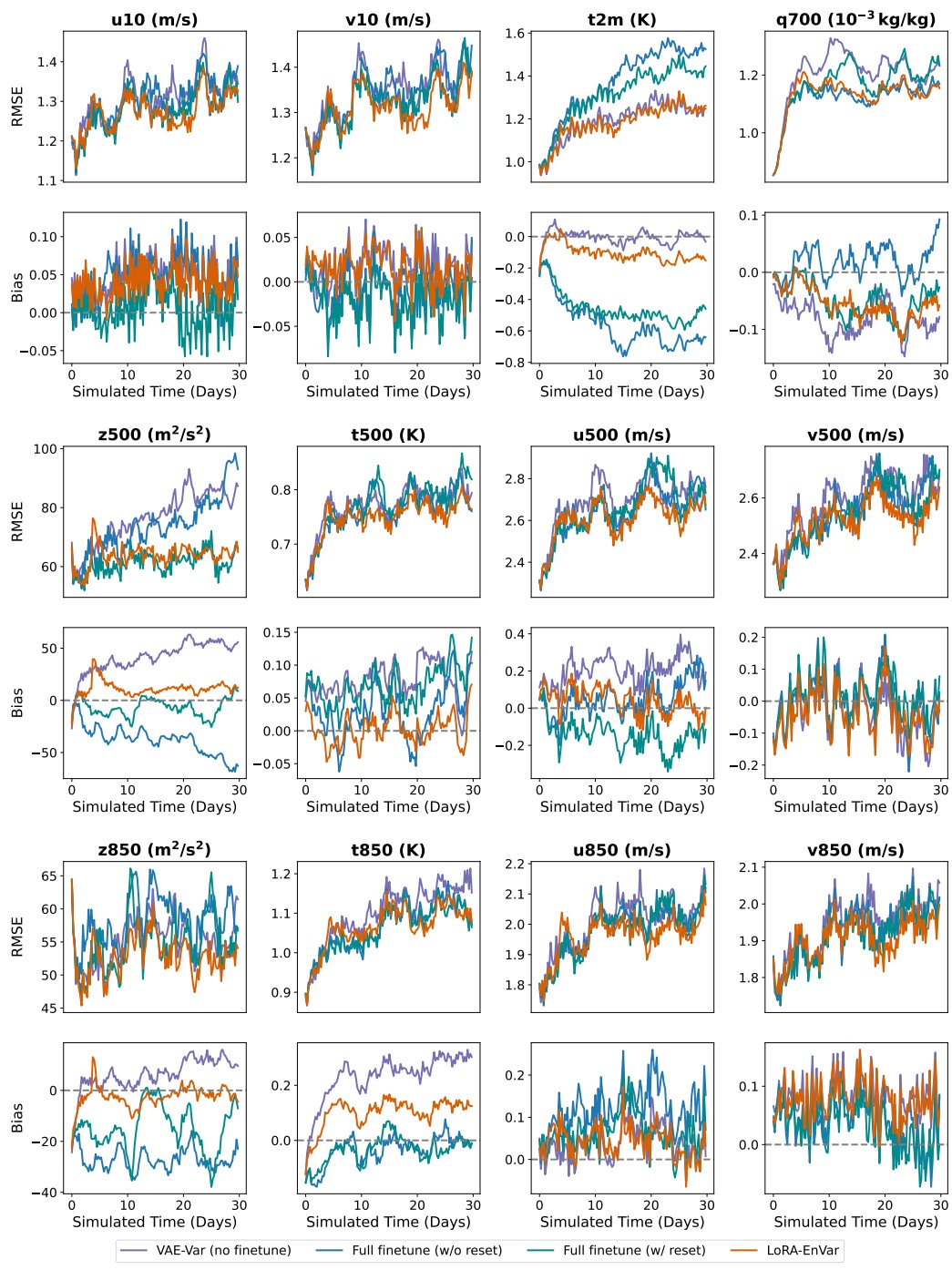

Figure 14: **Comparison of LoRA-EnVar and full finetuning methods using the 20-member ensemble.** LoRA-EnVar remains competitive while using significantly fewer trainable parameters.

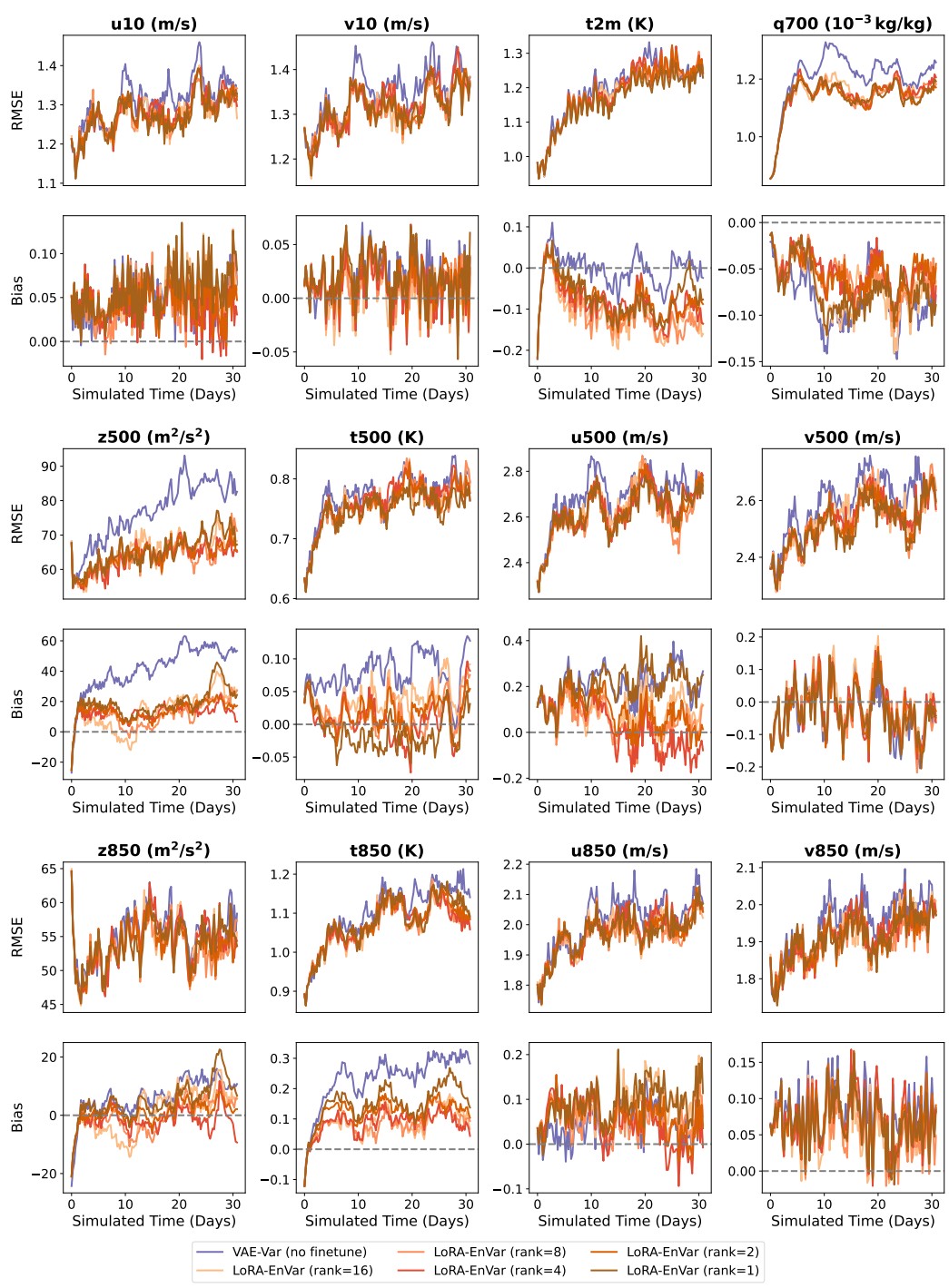

Figure 15: **RMSE and bias comparisons of LoRA-EnVar with different LoRA ranks (1, 2, 4, 8, 16).** Rank 2 offers a good trade-off between accuracy and efficiency, while rank 1 underperforms slightly on some variables.

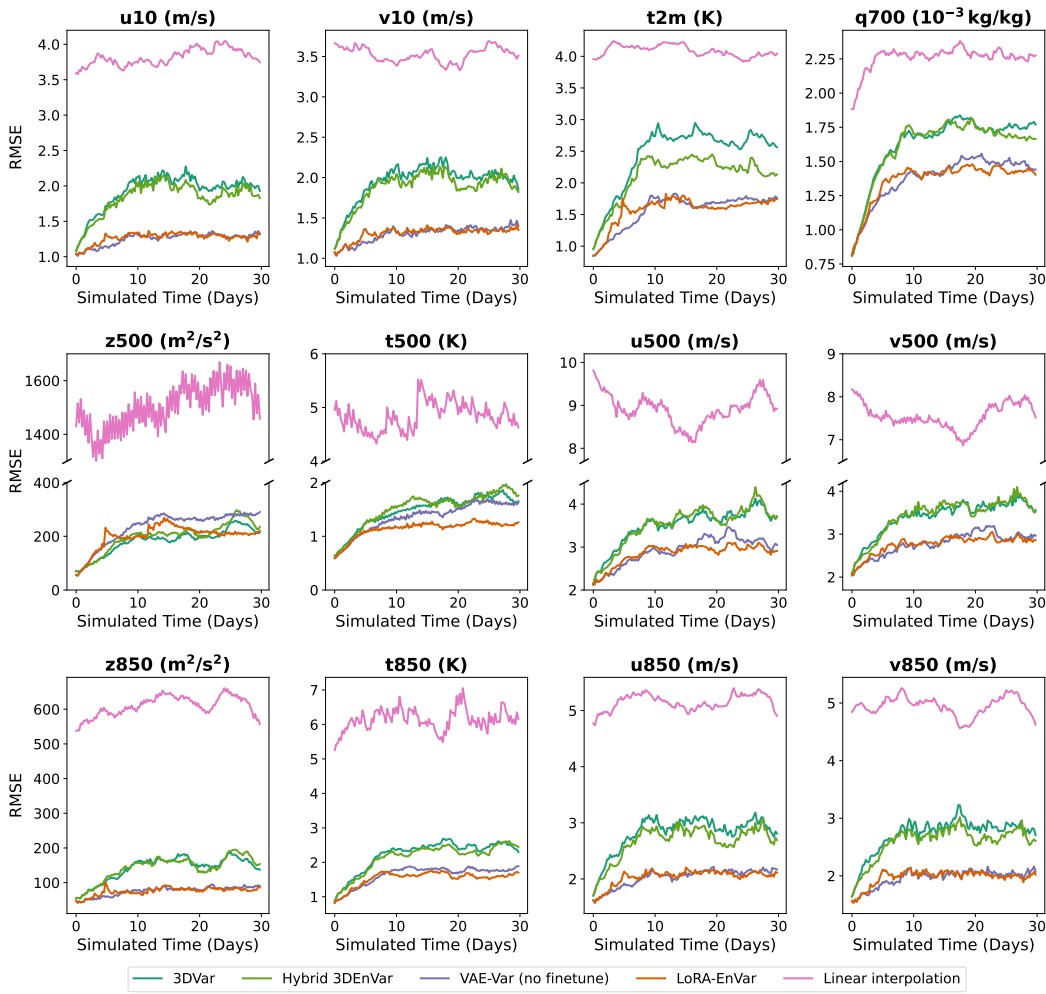

Figure 16: **ERA5-based evaluation of analysis RMSE for various methods using GDAS real observations.** All model-based assimilation approaches outperform linear interpolation, with LoRA-EnVar achieving the best overall accuracy.

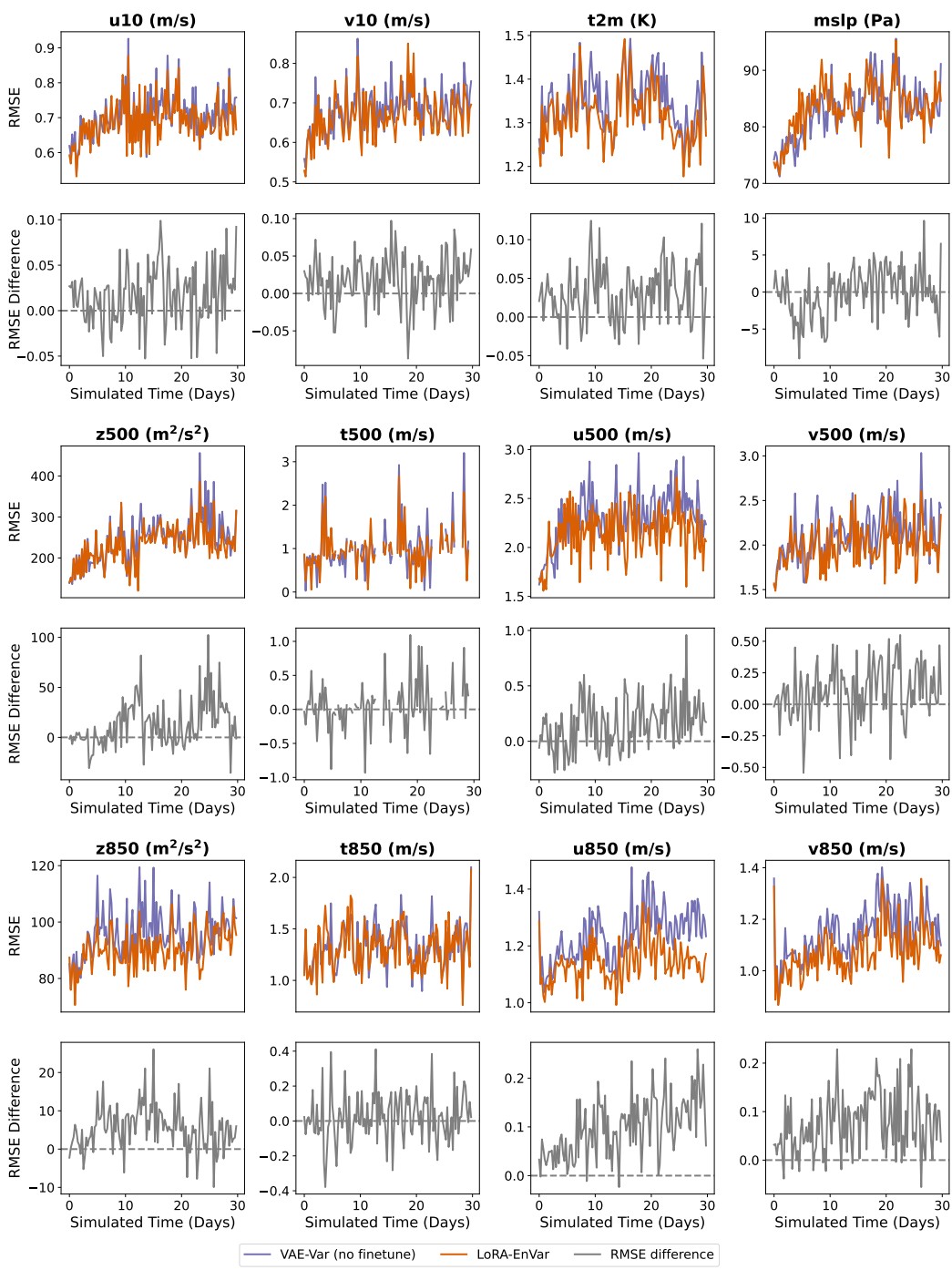

Figure 17: **Station-based evaluation using held-out GDAS observations.** The bottom half shows RMSE differences between LoRA-EnVar and VAE-Var, with positive values indicating better performance by LoRA-EnVar.

