# OpenReview forum: "LoRA-EnVar: Parameter-Efficient Hybrid Ensemble Variational Assimilation for Weather Forecasting"
_NeurIPS.cc/2025/Conference — NeurIPS 2025 poster_

### Official Review · Reviewer_pe2M · 2025-06-06

**Clarity:** 4
**Significance:** 4
**Originality:** 3
**Rating:** 6
**Confidence:** 4

**Summary:**

The authors develop a new variational assimilation method to support numerical weather prediction. First, a VAE is trained on historical forecast errors to serve as a baseline. Then, LoRA modules are inserted into the decoder. These LoRA modules are tuned at each time point using uncertainty information from AI-based ensemble forecasts. The resulting framework is intended to efficiently incorporate new information while keeping the historical background of the model.

**Questions:**

The space/time summaries of forecast errors seem to favor your method, does this still hold up when reduced to a single spatial/temporal mean? Appears to be the case but it’s not always clear to me (especially in parts of Figure 3).

**Ethical Concerns:**

["NO or VERY MINOR ethics concerns only"]

**Final Justification:**

This work adds the novel contribution of LoRA modules to the VAE-VAR architecture for climate data assimilation. A comprehensive spatiotemporal evaluation shows that the model offers a clear benefit for an important application area.

**Limitations:**

Yes.

**Quality:**

4

**Strengths And Weaknesses:**

Strengths: The online training of LoRA modules in the decoder serves a double purpose to vastly improve efficiency over full fine tuning while preserving the decoder’s original parameters. Empirical results show strong improvement over previous approaches in a vitally important application.

Weaknesses: While the included spatial and temporal summaries are of greater interest and importance, it may also be interesting to see overall accuracy metrics to provide full context (perhaps just for your main results). While the incorporation of LoRA modules is novel, the overall VAE approach to data assimilation is familiar.

---

> ### Author Rebuttal · Authors · 2025-07-26
>
> Thank you very much for your thoughtful and encouraging review. We are glad that you found the proposed approach and its evaluation to be of high quality and significance.
>
> ___
>
> > **Response to Q1&W1: Spatial/temporal mean for overall evaluation**
>
> In response to your helpful suggestion, we have included overall accuracy metrics in the table below, where each entry represents the spatial-temporal mean RMSE for a key variable (corresponding to Figure 3&4 in the main text and Figure 8&9 in the Appendix). **Bold** values indicate the best performance, and *italic* values indicate the second-best. As shown, LoRA-EnVar consistently ranks either first or second, closely matching or outperforming both Full Finetune (w/ reset) and Hybrid 3DEnVar, depending on the variable. This further confirms that the improvements observed in the space/time summaries also hold under global scalar evaluation.
>
> | Overall RMSE                            | 3DVar   | Hybrid 3DEnVar      | LoRA-EnVar          | VAE-Var (no finetune) | Full finetune (w/o reset) | Full finetune (w/ reset) |
> | --------------------------------------- | ------- | ------------------- | ------------------- | --------------------- | ------------------------- | ------------------------ |
> | z500 ($\mathrm{m}^2/\mathrm{s}^2$)      | $98.17$ | $84.79$             | $\underline{64.11}$ | $76.75$               | $94.93$                   | $\mathbf{61.68}$         |
> | z850 ($\mathrm{m}^2/\mathrm{s}^2$)      | $58.23$ | $54.92$             | $\underline{53.58}$ | $55.10$               | $67.98$                   | $\mathbf{53.56}$         |
> | u500 ($\mathrm{m}/\mathrm{s}$)          | $2.811$ | $\underline{2.681}$ | $\mathbf{2.623}$    | $2.691$               | $2.754$                   | $2.708$                  |
> | u850 ($\mathrm{m}/\mathrm{s}$)          | $2.272$ | $2.103$             | $\mathbf{1.971}$    | $2.011$               | $2.037$                   | $\underline{1.972}$      |
> | v500 ($\mathrm{m}/\mathrm{s}$)          | $2.786$ | $2.642$             | $\mathbf{2.523}$    | $2.592$               | $2.622$                   | $\underline{2.549}$      |
> | v850 ($\mathrm{m}/\mathrm{s}$)          | $2.247$ | $2.080$             | $\mathbf{1.908}$    | $1.948$               | $1.959$                   | $\mathbf{1.908}$         |
> | t500 ($\mathrm{K}$)                     | $0.757$ | $\mathbf{0.728}$    | $\underline{0.755}$ | $0.771$               | $0.793$                   | $0.776$                  |
> | t850 ($\mathrm{K}$)                     | $1.136$ | $\underline{1.075}$ | $\underline{1.075}$ | $1.101$               | $1.103$                   | $\mathbf{1.064}$         |
> | u10 ($\mathrm{m}/\mathrm{s}$)           | $1.539$ | $1.449$             | $\underline{1.275}$ | $1.308$               | $1.332$                   | $\mathbf{1.268}$         |
> | v10 ($\mathrm{m}/\mathrm{s}$)           | $1.593$ | $1.507$             | $\mathbf{1.314}$    | $1.347$               | $1.377$                   | $\underline{1.315}$      |
> | t2m ($\mathrm{K}$)                      | $1.499$ | $1.347$             | $\mathbf{1.179}$    | $1.194$               | $1.468$                   | $\underline{1.269}$      |
> | q700 ($10^{-3}\mathrm{kg}/\mathrm{kg}$) | $1.146$ | $\mathbf{1.129}$    | $\underline{1.136}$ | $1.208$               | $1.166$                   | $1.152$                  |
>
> > **Response to W1: Novelty of the VAE approach**
>
> Regarding your note on the familiarity of the VAE-based approach, we fully agree that the use of VAEs in data assimilation is established. Our contribution focuses on incorporating LoRA modules for efficient online adaptation, which we believe adds a novel and practically valuable enhancement to this VAE-Var framework.
>
> ___
>
>
> We appreciate your feedback once again and it helped us strengthen the overall presentation and completeness of the paper.

---

> > ### Comment · Reviewer_pe2M · 2025-08-02
> >
> > Thank you for the condensed summary table, I think this is a good supplement to the figures in the paper.

---

> > > ### Author Response · Authors · 2025-08-03
> > >
> > > Thank you for the recognition of our work. We’ll add these results in the revised version.

---

### Official Review · Reviewer_scAN · 2025-06-08

**Clarity:** 3
**Significance:** 3
**Originality:** 3
**Rating:** 4
**Confidence:** 3

**Summary:**

This work addresses the problem of data assimilation by proposing a data-driven method based on the VAE-VAR architecture. A key contribution is the explicit separation of two types of model error: climatological error and flow-dependent error. These are handled differently: the former through offline training, and the latter via online fine-tuning of a small subset of network weights using the LoRA (Low-Rank Adaptation) technique. Experimental results show the effectiveness of both online adaptation in general, and the LoRA-based method in particular, compared to other fine-tuning approaches.

If I understood correctly, the core components of the proposed method are VAE-VAR and LoRA. The main novelty compared to existing machine learning-based approaches lies in the ability to adapt the prior model online through LoRA.

**Questions:**

1. How does the proposed model compare with other architectures such as DiffDA? In line 45, the authors state: “but they do not incorporate climatological priors or flow-dependent ensembles.” Could the authors elaborate on this distinction and explain this mathematically, and how it translate into architectural or algorithmic differences?

2. The training procedure uses the NMC method (mentioned in line 59 and described in the Appendix). Could the authors clarify how training error samples are constructed using this method, and how these samples capture 'climatological' rather than 'flow-dependent' error?

3. Could the authors discuss the computational cost of their method? For instance:
   - Does the VAE structure reduce cost by enabling assimilation in latent space?
   - How does the LoRA-based online tuning compare in efficiency to other fine-tuning strategies?

**Ethical Concerns:**

["NO or VERY MINOR ethics concerns only"]

**Final Justification:**

The idea of an adaptive neural network for data assimilation is neat and original. While the rebuttal clarified most the clarity issues, the paper still lacks a proper mathematical formulation for the tackled problem. Therefore I have increased my score from 3 to 4.

**Limitations:**

Yes

**Quality:**

3

**Strengths And Weaknesses:**

Overall, the paper is clearly written and the idea is original. However, I found some clarity issues and a general lack of context.

### Strengths

- Data assimilation is a crucial challenge, and leveraging deep learning has the potential to significantly improve weather and climate forecasting.
- The paper is clearly written and easy to follow.
- The proposed technique allows low-cost parameter adaptation.
- The idea of adapting neural network parameters online is novel in this domain and appears both intuitive and powerful.
- The experimental results are promising and support the main claims.

### Weaknesses

1. The paper focuses exclusively on spatial data assimilation, where observations are assimilated once per time step. There is no mention of 4D-Var, which is the state-of-the-art approach and would be a natural point of comparison.

2. The distinction between climatological and flow-dependent error is not clearly defined. From my understanding, both arise from model forecast errors but differ in their time scales. Since this conceptual separation underpins the method, it would be valuable to define both types of error mathematically, for example, as expectations over model trajectories, and to clarify how they differ in structure and treatment.

3. The paper lacks technical depth in describing its main components, VAE-VAR and LoRA. While the plain English explanations are helpful, the mathematical formulation is minimal. VAE-VAR is central to the model, yet it is only briefly described, with most details deferred to Appendix A. Similarly, LoRA is treated as well-known, which may not be the case in the scientific machine learning community. The problem of efficiently adapting a parametric model that defines a distribution, given samples from that distribution, is nontrivial and deserves formal treatment. This could benefit from connections to frameworks like online learning, adaptive neural networks, and meta-learning. While much of the existing literature in these areas focuses on prediction and forecasting tasks, rather than on estimating prior distributions, there are strong conceptual overlaps regarding online model adaptation. In particular, recent works on low-rank meta-learning architectures for physical systems [1, 2] may offer useful insights and parallels.

4. Relatedly, the comparison to existing machine learning-based data assimilation methods (e.g., DiffDA) is somewhat superficial. A clearer description of the proposed architecture and its treatment of model error would enable more precise comparisons to alternative approaches. See also my Question 1.

5. In the paragraph beginning at line 103, no references are provided for traditional hybrid EnVar approaches. A citation would strengthen the context and situate the contribution within established literature.

### References

[1] Serrano, L., Koupaï, A. K., Wang, T. X., ERBACHER, P., & Gallinari, P. Zebra: In-Context and Generative Pretraining for Solving Parametric PDEs.

[2] Blanke, M., & Lelarge, M. Interpretable Meta-Learning of Physical Systems. In The Twelfth International Conference on Learning Representations.

---

> ### Author Rebuttal · Authors · 2025-07-30
>
> Thank you very much for your constructive and detailed comments. We greatly appreciate your positive assessment of the novelty and overall presentation of our work.
>
> ___
>
> > **Response to Q1&W4: comparison with other architectures such as DiffDA**
>
> By stating that "they do not incorporate climatological priors or flow-dependent ensembles," we mean that SDA does neither, while DiffDA incorporates a learned climatological prior but does not leverage flow-dependent ensembles.
>
> More formally, data assimilation aims to estimate the posterior $p(x^i \mid x_b^i, y^i)$, given the prior state $x_b^i$ and observations $y^i$ at time step $i$. One common approximation is to replace the time-specific distribution with a time-averaged conditional prior $p(x \mid x_b)$, which encodes climatological structure. DiffDA follows this approach and uses a diffusion model to learn a fixed mapping from $x_b$ to $x$, approximating $p(x \mid x_b)$. However, this mapping is static and cannot reflect the time-varying nature of $p(x^i \mid x_b^i)$ under dynamically evolving conditions.
>
> In contrast, VAE-Var, inspired by traditional variational data assimilation methods, assumes the forecast error $e^i = x^i - x_b^i$ is independent of $x_b^i$ and instead learns a time-averaged climatological error distribution $p(e)$. LoRA-EnVar builds upon this by incorporating flow-dependent adjustments to the error distribution. Specifically, it constructs online error samples using time-lagged forecasts and encodes them through low-rank updates, allowing the model to approximate time-varying distributions $p_i(e^i)$ and better capture regime-dependent uncertainties (e.g., stronger jet activity in winter).
>
> Importantly, this modeling choice has architectural and practical implications. DiffDA requires paired data $(x_b^i, x^i)$ to train its diffusion model, which can be difficult or even infeasible to obtain in online or operational settings. In contrast, LoRA-EnVar only requires access to error samples $e^i$, which can be easily estimated from ensembles using established methods such as time-lagging. Therefore, LoRA-EnVar offers greater flexibility and is better suited for real-time deployment.
>
> > **Response to Q2: How training error samples are constructed using the NMC method**
>
> To construct training error samples using the NMC method, we first select historical ERA5 reanalysis states at 12-hour intervals. For each selected state, we run the forecast model to generate two forecasts for the same target time: one with a 12-hour lead time and another with a 24-hour lead time. The difference between these two forecasts forms an error sample. This process is repeated over a 37-year span of ERA5 data, resulting in a large and diverse training set. Because the errors are aggregated over many years and meteorological regimes, the learned distribution reflects averaged, climatological characteristics rather than case-specific, flow-dependent variations. In contrast, the samples used during finetuning are constructed online from recent ensemble forecasts and are therefore sensitive to current atmospheric conditions. These are referred to as “flow-dependent” errors in our framework.
>
> > **Response to Q3: Discussion on the computational cost**
>
> Thank you for the question.
>
> - The primary benefit of VAE-Var over traditional 3DVar is its ability to capture non-Gaussian error structures via a learned latent representation. While 3DVar applies a linear transformation to a Gaussian control variable, VAE-Var maps a latent variable to physical space using a neural network decoder. As such, the computational complexity of VAE-Var is not lower; in fact, it involves more floating-point operations. However, VAE-Var, implemented with optimized neural network operators, is highly compute-efficient on GPUs. In our experiments, one assimilation cycle using VAE-Var takes approximately 18 seconds on a single NVIDIA A100 GPU, compared to about 4 minutes per cycle using a CPU-based 3DVar implementation running on 16 cores.
> - Although LoRA updates only a small subset of parameters, the backward pass still traverses the entire network to compute gradients. In practice, LoRA reduces the fine-tuning time modestly, from ~10s per cycle in full fine-tuning to ~9s with LoRA. Nevertheless, The key advantage of LoRA lies not in time savings but in memory efficiency and modularity. Because LoRA modifies only a few low-rank matrices, the adapted model can be stored and transferred compactly, which is especially valuable when tracking adaptations over time or switching between flow regimes. Moreover, LoRA enables efficient online updates without altering the backbone, simplifing deployment and mitigating catastrophic forgetting.
>
> > **Response to W1: Comparison with 4DVar**
>
> To evaluate our framework under 4DVar settings, we constructed an extended observation scenario where, in addition to the standard 6-hourly observations (e.g., at 0h, 6h, 12h), we introduce intermediate observations at 1h, 7h, 13h, etc. We employ a 1-hour resolution version of the FengWu forecast model to resolve temporal dynamics within each 6-hour assimilation window.
>
> We compared eight methods and the results over the first 30 days of January are summarized in the tables below. As expected, all 4D variants outperform their 3D counterparts, reflecting the benefit of temporal assimilation. Importantly, LoRA-4DEnVar consistently outperforms 4DVar and Hybrid 4DEnVar, demonstrating both the extensibility and effectiveness of our framework in fully 4D assimilation settings.
>
> |z500 RMSE ($\mathrm{m^2}/\mathrm{s^2}$)|3DVar|Hybrid 3DEnVar|4DVar|Hybrid 4DEnVar|VAE-3DVar|LoRA-3DEnVar|VAE-4DVar|LoRA-4DEnVar|
> |-|-|-|-|-|-|-|-|-|
> |Day 1-10|80.434|80.937|70.331|72.894|66.724|60.334|60.609|54.200|
> |Day 11-20|86.261|88.631|74.870|78.942|77.401|64.602|71.187|59.024|
> |Day 21-30|85.746|84.153|72.278|76.135|85.516|67.135|73.638|61.156|
> |Average|84.147|83.820|72.493|75.990|76.547|64.024|68.478|58.127|
>
> |u500 RMSE ($\mathrm{m}/\mathrm{s}$)|3DVar|Hybrid 3DEnVar|4DVar|Hybrid 4DEnVar|VAE-3DVar|LoRA-3DEnVar|VAE-4DVar|LoRA-4DEnVar|
> |-|-|-|-|-|-|-|-|-|
> |Day 1-10|2.761|2.611|2.725|2.565|2.595|2.539|2.445|2.440|
> |Day 11-20|2.918|2.730|2.879|2.680|2.729|2.649|2.536|2.533|
> |Day 21-30|2.923|2.691|2.826|2.638|2.743|2.670|2.592|2.582|
> |Average|2.867|2.653|2.810|2.628|2.689|2.619|2.524|2.519|
>
> > **Response to W2: Mathematical definition of two types of error**
>
> We thank the reviewer for the valuable suggestion. To clarify the conceptual distinction, we provide a more precise mathematical formulation. We denote the forecast error at time step $i$ as $e^i = x^i - x_b^i$, where $x^i$ is the true state and $x_b^i$ is the background forecast. Then:
>
> - The **flow-dependent error distribution** refers to the conditional error distribution at a particular time $i$, given the background flow conditions $\mathcal{F}_i$ (e.g., the synoptic-scale state of the atmosphere): $p_i(e) := p(e^i \mid \mathcal{F}_i)$. This formulation acknowledges that the forecast error structure depends on the current dynamical regime. For instance, during periods with strong jet streams or tropical cyclones, the forecast uncertainty tends to be larger and more anisotropic.
> - The **climatological error distribution** is defined as the time-averaged distribution of forecast errors: $p_{\text{clim}}(e) := \mathbb{E}_i \, p_i(e)$. This captures the average statistical structure of forecast errors across a long historical period and is often used in traditional climatology-based assimilation schemes.
>
> While both types of error indeed arise from model forecast errors, we respectfully note that the distinction is not solely based on time scales. The key difference, particularly in the context of ensemble-based assimilation, lies in whether the error statistics are conditioned on the current flow state. Flow-dependent errors reflect state-specific uncertainties, while climatological errors represent the marginal distribution averaged over many flow regimes.
>
> > **Response to W3: Scope to strengthen mathematical formulation and conceptual links**
>
> Thank you for the thoughtful suggestion. While many recent works in the SciML community focus on general-purpose physical surrogates or PDE solvers, our work centers on improving probabilistic data assimilation in a large-scale, operational setting—namely, numerical weather prediction. Our adoption of LoRA is driven not only by considerations of efficiency, but also by the need for spatially localized, flow-dependent adaptation within a fixed VAE backbone. We agree that connecting our approach more clearly to the broader frameworks of online learning and meta-learning would enhance the conceptual framing of the paper. In particular, we appreciate the pointer to recent work on low-rank meta-learning architectures for physical systems, which indeed parallels our use of LoRA as a mechanism for cycle-wise adaptation. Although our primary motivation is practical—capturing flow-dependent uncertainty efficiently in high-dimensional systems—we recognize that the underlying problem structure closely aligns with meta-adaptation paradigms, where compact parameter updates enable generalization across evolving regimes. We will revise the manuscript to clarify these connections, formalize the VAE and LoRA components more precisely, and cite the suggested references accordingly.
>
> > **Response to W5: Missing Hybrid EnVar citation**
>
> Thank you for pointing this out. The implementation we adopt for Hybrid 3DEnVar follows the formulation described in [1]. We will include this citation in the revised version.
>
> ___
>
> We appreciate your feedback once again and it helped us strengthen the overall presentation and completeness of the paper.
>
> [1] Wang, X., Snyder, C., & Hamill, T. M. (2007). On the theoretical equivalence of differently proposed ensemble–3DVAR hybrid analysis schemes. *Monthly Weather Review*, *135*(1), 222-227.

---

> ### Comment · Reviewer_scAN · 2025-08-01
> **Response to the author's rebuttal**
>
> We thank the authors for their clarifications.
>
> The additional details provided in the rebuttal help clarify the method and would strengthen the paper if incorporated into a revised version. However, I still find that the lack of mathematical exposition of the core components, specifically the VAE-Var framework and the online distribution adaptation via LoRA, limits the clarity of the proposed approach.
>
> Besides, could the authors provide more details on the adaptation of LoRA-EnVar to 4DVar?

---

> > ### Author Response · Authors · 2025-08-02
> >
> > Thank you for the follow-up comments.
> >
> > We acknowledge that the current manuscript could benefit from a more rigorous mathematical exposition of the core components, particularly VAE-Var and LoRA-based online adaptation. In future versions, we will aim to make these formulations more precise and accessible in the main text. The additional details provided in the rebuttal help clarify the method and would strengthen the paper if incorporated into a revised version.
> >
> > Regarding the extension to 4DVar: While LoRA-EnVar is designed to capture flow-dependent background error covariance across assimilation windows, it does not explicitly account for the temporal distribution of observations within a single window. In operational settings, however, observations often arrive at sub-daily intervals, and ignoring their time structure may limit assimilation quality. To address this, LoRA-EnVar can be naturally extended to a 4D variational setting.
> >
> > In 4DVar, the observation loss is generalized to incorporate model evolution and temporally distributed observations:
> >
> > $\mathcal{L}\_o\left(\mathbf{x}, \\{\mathbf{y}\_i\\}\_{i=0}^{N-1}\right) = \frac{1}{2} \sum_{i=0}^{N-1} \left(\mathbf{y}_i - \mathcal{M}\_{0\to i}\left(\mathbf{x}\right)\right)^\mathrm{T} \mathbf{R}^{-1} \left(\mathbf{y}\_i - \mathcal{M}\_{0\to i}\left(\mathbf{x}\right)\right),$
> >
> > where $\mathcal{M}_{0\to i}$ denotes the model forecast from initial time to step $i$. When the model and observation operators are differentiable, this term is fully compatible with latent-space optimization. Accordingly, we replace the observation term in LoRA-EnVar with:
> >
> > $\tilde{\mathcal{L}}_o(\mathbf{z}) = \mathcal{L}_o(\mathcal{D}_j(\mathbf{z}) + \mathbf{x}_b, \\{\mathbf{y}\_i\\}\_{i=0}^{N-1}),$
> >
> > where $\mathcal{D}_j$ is the decoder and $\mathbf{x}_b$ the background state, with the subscript $j$ indicating the $j$th assimilation window. The decoder $\mathcal{D}_j$ includes a LoRA module designed to adapt to changes in background error statistics across assimilation windows (as opposed to within a single window). Optimization proceeds in the same way as in the 3D case.
> >
> > In our additional experiments, we applied this 4DVar extension to both hybrid 4DEnVar and LoRA-EnVar. Each assimilation window incorporated two observation times spaced one hour apart (i.e., $N=2$), demonstrating that LoRA-EnVar remains effective and compatible in temporally-resolved assimilation settings.
> >
> > We welcome any further feedbacks or suggestions.

---

> > > ### Comment · Reviewer_scAN · 2025-08-03
> > >
> > > Thank you for the additional details. This clarified the potential application of the proposed method in a spatio-temporal data assimilation settings.
> > >
> > > While  the rebuttal improved the clarity of the paper, I decide to maintain my score for the reasons I've mentioned above.

---

> > > > ### Author Response · Authors · 2025-08-03
> > > >
> > > > Thank you for the feedback. We’ll reflect the clarifications in the revised version.

---

### Official Review · Reviewer_Zgna · 2025-06-15

**Clarity:** 3
**Significance:** 3
**Originality:** 4
**Rating:** 5
**Confidence:** 4

**Summary:**

The paper aims to address limitations in current variational data assimilation (DA) systems, most notably, their reliance on using a static background error learned from climatological data, and the limitation to using Gaussian errors to generate analysis. While hybrid DA methods exist that combine background error statistics with "error-of-the-day" statistics obtained by ensemble simulations, traditionally, this still assumes the error to be Gaussian to be used within the traditional variational framework. The paper proposes an algorithm that performs variational data assimilation in latent space using VAEs trained on climatological error samples; assuming a Gaussian bottleneck for the VAE, the use of Gaussian error statistics is justified in latent space by construction. Furthermore, the VAE is fine-tuned using LoRA on "errors-of-the-day" to achieve a dynamically evolving error statistics. Experiments on a weather forecasting model trained on ERA5 demonstrate the added benefits of using non-Gaussian error estimates (achieved by operating in latent space) and dynamically evolving error by LoRA finetuning.

**Questions:**

- How are the ensemble obtained to fine-tune LoRA (i.e., what are the "Other predictions" in Figure 1)? Are they based on different initial conditions or different forecasting models?

**Ethical Concerns:**

["NO or VERY MINOR ethics concerns only"]

**Final Justification:**

The methodology introduced by the paper is, in my opinion, quite an interesting extension to classical hybrid ensemble variational methods, which, traditionally, are limited to using Gaussian error covariances. I found the experiments to be well-conducted and presented, although Reviewer 2YwJ points out the possibility that it may be flawed due to concerns about how the ensembles are calibrated. Nonetheless, I find it to be an interesting direction in data assimilation with potential for further improvements down the line, which is why I would recommend an acceptance.

**Limitations:**

The authors acknowledge limitations of the approach, in particular, lack of theoretical characterisation and intrinsic limitation of only being able to produce point estimates.

**Paper Formatting Concerns:**

There are no paper formatting concerns as far as I can see.

**Quality:**

3

**Strengths And Weaknesses:**

__Strengths:__
- Overall, I believe that the idea presented in this work is very elegant, providing an interesting approach to tackle both the issues of using static and Gaussian error covariances in traditional data assimilation using a single clean framework.
- The experiment section is quite strong. In particular, the ablation studies convincingly demonstrate the added benefit of using both non-Gaussian error statistics and dynamical error statistics in data assimilation, at different observation densities.
- Experiments to compare LoRA vs full fine-tuning demonstrate that we are not losing much by only fine-tuning the VAE; in fact, the experiments show that LoRA is not only sufficient, but even outperforms the results of full fine-tuning. in general, I believe that all of the components that are introduced in the model is justified.

__Weaknesses:__
- One clear limitation of the approach is that the method explicitly uses the fact that the VAE has a Gaussian bottleneck to justify the use of 3DVAR in latent space. Thus, it cannot accommodate the use of more general autoencoders such as vector-quantised VAEs (VQ-VAEs), which can achieve better reconstruction compared to standard VAEs.
- A minor gripe that I have is that some of the colors used in the plots are very similar to each other, which makes it hard to see which curve corresponds to which model.
- I believe the proposed methodology can be better explained. For example, I believe it would be useful to have Appendix A in the main body instead of the appendix. Otherwise, we are only left with a vague description of the methodology and Figure 1 to understand the idea, which I initially found difficult to grasp until reading Appendix A.

---

> ### Author Rebuttal · Authors · 2025-07-26
>
> Thank you very much for your constructive and detailed comments. We greatly appreciate your positive assessment of the overall methodology and experimental evaluation.
>
> ___
>
> > **Response to Q1: How the ensemble is obtained to finetune LoRA**
>
> Thank you for the question regarding the source of ensemble forecasts used to fine-tune the LoRA modules (“Other predictions” in Figure 1). The LoRA-EnVar framework is flexible and can support different strategies for ensemble forecast generation, such as perturbing initial conditions or using multiple forecast models. In our current implementation, we adopt a time-lagged ensemble approach, which is well adopted in use and fundamentally falls into the first category—based on different initial conditions. Specifically, all ensemble members are generated using the same forecast model (FengWu), but from earlier analysis times with different lead times (e.g., 1, 2, or 3 steps), such that they all arrive at the same target time. This strategy offers a computationally efficient way to represent flow-dependent uncertainty without requiring multiple models or explicit perturbations. We have provided the detailed implementation of this scheme in Appendix C.2, and the ensemble structure is illustrated in Figure 7.
>
> > **Response to W1: Support for other autoencoder architectures**
>
> We fully agree that our current method relies on the assumption of a Gaussian latent space, which underpins the validity of applying 3DVar in latent space. While LoRA can be incorporated into more expressive autoencoders such as VQ-VAEs, particularly those based on Transformer architectures, we have not yet developed a data assimilation framework that is compatible with vector-quantized latent representations. Unlike VAEs with continuous latent spaces, the discrete nature of VQ-VAEs poses unique challenges when integrated with variational assimilation, including the absence of gradient flow in the discrete latent space and the difficulty of formulating background error priors. We will explicitly clarify this limitation and consider this an important direction for future work in the revised version.
>
> > **Response to W2: Colors used in the plots**
>
> Thank you for pointing this out. We acknowledge that some color choices in the figures may make it difficult to distinguish between different curves. In the revised version, we will improve the figure design by optimizing the color palette and line styles to enhance clarity and accessibility.
>
> > **Response to W3: Better explanation of the proposed methodology**
>
> We appreciate your suggestion to improve the presentation of the methodology. In the revised version, we will move the description of VAE-Var (currently in Appendix A) into the main text, so that the algorithmic pipeline can be more easily understood without requiring readers to consult the appendix.
>
> ___
>
> We appreciate your feedback once again and it helped us strengthen the overall presentation and completeness of the paper.

---

> > ### Comment · Reviewer_Zgna · 2025-08-03
> > **Thank you for the clarifications**
> >
> > Thank you for the overall clarifications. Upon reading the other reviews and the responses, I would like to maintain my positive support for this work. The proposed framework is a clever extension of classical Hybrid filters that makes great use of recent developments in machine learning, with extensive experiments (including the additional ones conducted during the rebuttal) backing up the advantages brought by the framework. However, raising the score to a 6 would require the paper to have a "groundbreaking impact on one or more areas of AI", hence, I will maintain my current score of 5.

---

> > > ### Author Response · Authors · 2025-08-03
> > >
> > > Thank you for the positive support and thoughtful assessment. We appreciate your encouraging feedback.

---

### Official Review · Reviewer_2YwJ · 2025-06-30

**Clarity:** 3
**Significance:** 2
**Originality:** 2
**Rating:** 3
**Confidence:** 4

**Summary:**

The submission introduces a new data assimilation approach which uses a pretrained VAE to learn an approximation of a non-Gaussian background covariance which is then updated online via LoRA in order to incorporate information from the local flow. This approach is empirically validated against several traditional approaches and the recently proposed VAE-Var on both assimilation from reanalysis ICs and simulated operational ICs across a variety of observation densities.

**Questions:**

Q1 - How does the calibration of the ensembles resulting from this approach compare to those of the competitive approaches?

Q2 - It is often easy to reduce RMSE by reducing high-frequency details and extremes in the forecast. Does the distribution of the resulting field values match the target?

Q3 - Does this significant competitive advantage persist when compared against modern DA tools?

**Ethical Concerns:**

["NO or VERY MINOR ethics concerns only"]

**Final Justification:**

On first read, my primary concern was the extent of validation. Including only one month of data and only a trivial baseline made it difficult to place the paper in context.

However, discussion seemed to bring up more issues. The ensembles were generally poorly calibrated and the inclusion of 21st century baselines produced results that did not match up with accepted knowledge in the domain. The authors were unable to produce satisfying explanations for these inconsistencies and it on deeper read, it seems experimental flaws are the most likely culprit.

While I think the paper on the whole is well motivated and contains many interesting ideas, I am uncomfortable recommending acceptance given the issues identified in the experimental setup and the lack of discussion of these issues in the submission and rebuttal.

**Limitations:**

Negative societal impact yes. However, discussion of limitations largely focuses on trivial limitations without discussing limitation in experimental methodology.

**Quality:**

3

**Strengths And Weaknesses:**

Strengths:

- S1 - The challenge of efficiently capturing non-Gaussian variation in data assimilation is a key challenge in modern weather forecasting.
- S2 - The combination of experiments progressively challenges the model. The use of restricted observations in space and time are very important and relevant to practical usage and greatly strengthen the potential impact.
- S3 - The extensive ablations are really interesting and provide some somewhat counter-intuitive information (LoRA providing the biggest gains for slower moving fields like Z500 in Fig 3, larger amounts of finetunable parameters hurting performance). This could be explored more strengthening the paper.
- S4 - Performance at smaller ensemble sizes is extremely promising. In an operation setting this could significantly reduce the cost of the forecast.

Weaknesses:
- W1 - The major weaknesses of the paper are centered on the evaluation and comparisons.
    - W1.A - The comparisons in general are very weak. The proposed method utilizes a 300M parameter SWINv2 architecture while 3DVar is largely a teaching example. This is not a fair comparison. While it's not expected that the experiments use a full operational DA system, a variational-to-variational comparison should include 4DVar (and associated ensemble-equipped methods) and likely at least one EnKF-based method (as they tend to be a bit easier to tune). Given the small ensemble sizes, it's likely that these would require some parameter tuning.
    - W1.B - On the ML side, the related work discussed a number of different approaches while stating without evidence why these approaches are unsuitable. This is a largely empirical submission, so these claims need to be shown rather than just stated. If scale is an issue, it's fine to just point to an appendix section showing why those approaches shouldn't scale. If the issue is instead that methods perform poorly at scale, then this can be shown experimentally.
    - W1.C - Right now the paper is motivated by fixing an error in the probabilistic formulation of variational DA. However, there is then no follow-up on calibration of forecasts. The paper states that this is not available for variational methods which is true for pure variational approaches where all spread information is provided a priori, but with hybrid methods, this can be estimated empirically from the ensemble.
 - W2 - All experiments are currently performed on a fairly small time window. 4.2 is on January 1, 2022. 4.3 is on that same month. There does not seem to be a reason why more extensive backtesting would be impossible.
- W3 - (Minor) Parameter efficiency is interesting, but not really the relevant comparison given your experiments show LoRA outperforming full rank training. Since online training is involved, LoRA should actually provide some real hardware efficiency gains either on runtime for fixed hardware or memory usage. These are more relevant.
- W4 - (Minor) For cases where a term or definition is vital to understanding the paper, it's usually better to find a way to include it in the main text rather than pointing readers to citations or the appendix. The variational procedure is the most notable case of this in the text currently.

Overall, this feels quite close to the line. Maintaining results with nontrivial numerics baselines and performing more extensive testing across time is likely enough that I'd be comfortable moving to weak accept while including stronger ML baselines, including analysis of the field distributions (appendix is fine as long as it's pointed out and discussed in the main body), and more probabilistic analysis would result in a stronger evaluation.

---

> ### Author Rebuttal · Authors · 2025-07-30
>
> Thank you very much for your constructive and detailed comments. We greatly appreciate your positive assessment of the methodology of our work.
>
> ___
>
> > **Response to W1.A&W1.B&Q3: Comparison with modern DA tools**
>
> **Comparison with traditional DA methods**
>
> We fully agree that stronger baselines, such as 4DVar and EnKF-based methods, are essential for a meaningful evaluation. In response, we extended our experiments to include:
>
> - **4DVar and Hybrid 4DEnVar:** To evaluate 4D variants, we introduce additional observations at 1h, 7h, 13h, etc., alongside standard 6-hourly observations. We use a 1-hour-resolution FengWu model to represent flow evolution within each assimilation window.
> - **LETKF:** We implemented the Local Ensemble Transform Kalman Filter with 200 km horizontal localization and 0.3 vertical tapering. Ensemble size was kept consistent with other EnVar-based methods.
>
> **Comparison with learning-based DA methods**
>
> We carefully considered several recent ML-based DA baselines:
>
> - **DiffDA** learns $p(x \mid x_b)$ via diffusion models, which requires paired $(x_b, x)$ data for training and adaptation. However, such labeled data is difficult to obtain online, making flow-dependent adaptation nontrivial. Our method instead models the error distribution $p(e = x - x_b)$, which is easier to sample in operational settings. We nonetheless include DiffDA (without flow-dependent ensembles) in our comparisons.
> - **SDA** does not incorporate $x_b$ and cannot perform cyclical assimilation, making it incompatible with our setting.
> - **FuXi-En4DVar** is a variant of Hybrid 4DEnVar with FuXi used for ensemble generation. We include a comparable Hybrid 4DEnVar baseline using FengWu.
> - **EnSF and Latent-EnSF**: EnSF assumes complete observations and is not applicable to high-dimensional sparse settings. Latent-EnSF extends EnSF to sparse observations, but its results are only shown at low spatial resolution ($21 \times 72 \times 144$), and no official code is currently available for reproduction or scaling tests. We will include it in future work once the implementation becomes accessible.
>
> **Results**
>
> In the updated tables below, we report RMSEs for key variables (z500 and u500) across **ten methods**, including five traditional and five learning-based approaches (including both 3D and 4D variants). These comparisons demonstrate that our proposed LoRA-EnVar performs competitively against both advanced DA systems and state-of-the-art ML-based approaches.
>
> | z500 RMSE ($\mathrm{m^2}/\mathrm{s^2}$) | 3DVar  | Hybrid 3DEnVar | 4DVar  | Hybrid 4DEnVar | LETKF | VAE-3DVar | LoRA-3DEnVar | VAE-4DVar | LoRA-4DEnVar | DiffDA |
> | - | - | - | - | - | - | - | - | - | - | - |
> | Day 1-10 | 80.434 | 80.937 | 70.331 | 72.894 | 102.83 | 66.724    | 60.334       | 60.609    | 54.200 | 125.12 |
> | Day 11-20 | 86.261 | 88.631 | 74.870 | 78.942 | 106.99 | 77.401    | 64.602       | 71.187    | 59.024 | 140.40 |
> | Day 21-30 | 85.746 | 84.153 | 72.278 | 76.135 | 119.77 | 85.516    | 67.135       | 73.638    | 61.156 | 149.41 |
> | Average | 84.147 | 83.820 | 72.493 | 75.990 | 109.86 | 76.547    | 64.024       | 68.478    | 58.127 | 138.31 |
>
> | u500 RMSE ($\mathrm{m}/\mathrm{s}$) | 3DVar | Hybrid 3DEnVar | 4DVar | Hybrid 4DEnVar | LETKF | VAE-3DVar | LoRA-3DEnVar | VAE-4DVar | LoRA-4DEnVar | DiffDA |
> | - | - | - | - | - | - | - | - | - | - | - |
> | Day 1-10 | 2.761 | 2.611 | 2.725 | 2.565 | 2.773 | 2.595     | 2.539        | 2.445     | 2.440 | 3.104 |
> | Day 11-20 | 2.918 | 2.730 | 2.879 | 2.680 | 2.518 | 2.729     | 2.649        | 2.536     | 2.533 | 3.306 |
> | Day 21-30 | 2.923 | 2.691 | 2.826 | 2.638 | 2.497 | 2.743     | 2.670        | 2.592     | 2.582 | 3.354 |
> | Average | 2.867 | 2.653 | 2.810 | 2.628 | 2.596 | 2.689     | 2.619        | 2.524     | 2.519 | 3.255 |
>
> > **Response to W1.C&Q1: Analysis of the calibration of the ensembles**
>
> Thank you for raising this important point. Our current work focuses on improving the analysis given a background ensemble, and we did not initially emphasize calibration analysis. To address this, we now include a new experiment to evaluate ensemble calibration. We ran both LoRA-EnVar and Hybrid 3DEnVar systems starting from July 1st and assessed the ensemble forecast initialized at 00:00 on July 15th, allowing sufficient spin-up for both methods. The results are summarized below:
>
> | z500 | Hybrid 3DEnVar | LoRA-EnVar |
> | - | - | - |
> | RMSE | 113.50 | 82.20 |
> | Spread | 70.61 | 76.10 |
> | Spread-to-RMSE Ratio | 62.2% | 92.6% |
>
> | u500 | Hybrid 3DEnVar | LoRA-EnVar |
> | - | - | - |
> | RMSE | 2.840 | 2.587 |
> | Spread  | 1.423 | 1.385 |
> | Spread-to-RMSE Ratio | 50.1% | 53.5% |
>
> These results show that LoRA-EnVar not only improves forecast accuracy (lower RMSE), but also yields better-calibrated ensembles, as indicated by the higher spread-to-RMSE ratios. This suggests that the flow-dependent latent prior learned by LoRA-EnVar more effectively captures uncertainty in the analysis compared to traditional hybrid methods. We will include additional calibration diagnostics in the revised version to further support this finding.
>
> > **Response to W2: More extensive testing across time**
>
> Thank you for the suggestion. We fully agree that evaluating the system solely on January data is insufficient to demonstrate generalizability. In response, we have conducted an additional experiment using data from July, which represents a different seasonal regime.
>
> | z500 RMSE ($\mathrm{m^2}/\mathrm{s^2}$) | 3DVar | Hybrid 3DEnVar | LoRA-EnVar | VAE-Var (no finetune) | Full finetune (w/o reset) | Full finetune (w/ reset) |
> | - | - | - | - | - | - | - |
> | Day 1-10 | 84.310 | 85.128 | 62.146 | 70.980 | 93.606 | 59.597 |
> | Day 11-20 | 88.768 | 91.054 | 68.911 | 87.594 | 137.540 | 68.526 |
> | Day 21-30 | 87.170 | 87.920 | 73.072 | 87.623 | 130.110 | 71.267 |
> | Average | 86.749 | 88.034 | 68.043 | 82.066 | 120.419 | 66.463 |
>
> | u500 RMSE ($\mathrm{m}/\mathrm{s}$) | 3DVar | Hybrid 3DEnVar | LoRA-EnVar | VAE-Var (no finetune) | Full finetune (w/o reset) | Full finetune (w/ reset) |
> | - | - | - | - | - | - | - |
> | Day 1-10 | 2.725 | 2.581 | 2.514 | 2.550 | 2.598 | 2.489 |
> | Day 11-20 | 2.815 | 2.713 | 2.707 | 2.709 | 3.043 | 2.764 |
> | Day 21-30 | 2.827 | 2.713 | 2.747 | 2.715 | 2.898 | 2.824 |
> | Average | 2.789 | 2.669 | 2.656 | 2.658 | 2.846 | 2.692 |
>
> > **Response to W3: Hardware efficiency gains**
>
> Thank you for the comment. While LoRA does not significantly reduce runtime (from $\sim$10s to $\sim$9s per cycle) or memory (~40.5GB to ~39.1GB) compared to full-finetune methods, its main advantage lies in dramatically lower storage cost. A single LoRA checkpoint is only ~648KB, enabling long-term retention of fine-tuned states (e.g., ~2.5GB for one year at 6-hour cycles), compared to ~2.3TB for full fine-tuning, which is over 1000× smaller. This makes LoRA-EnVar much more practical for retrospective analysis or downstream applications.
>
> > **Response to W4: Key formulations not presented in main body**
>
> Thank you for the helpful suggestion. We agree that the variational procedure is important and should be briefly clarified in the main text. In variational data assimilation, the goal is to estimate the state that minimizes a cost function balancing prior and observational constraints, equivalent to a regularized negative log-posterior. In our case, this is formulated in a latent space with learned priors. We will add concise explanations in the revised version to improve clarity and self-containment.
>
> > **Response to Q2: Concerns about realism of forecast distribution**
>
> Thank you for raising this important point. Our LoRA-EnVar framework does not rely on reducing high-frequency features or extremes to lower RMSE. To examine whether our analysis field preserves realistic distributional characteristics, rather than artificially smoothing the output, we compute two standard statistical metrics: standard deviation and skewness, which reflect the overall spread and asymmetry of the field values, respectively.
>
> Due to limits on image uploads, we report numerical values at a representative time (2022.01.01 00:00) in the table below. It can be observed that:
>
> - The standard deviation of the analysis field is similar to than that of the background field, indicating that the assimilation process does not suppress variance.
> - The skewness values of both z500 and u500 are well aligned with those of the ground truth, suggesting that extreme values are retained, and the distributional shape is preserved.
>
> The relative differences between the analysis and the ground truth are below 1%, confirming that our method does not distort the statistical structure of the field. In the revised version, we will include additional visualizations and statistical analyses in the appendix to further support this point.
>
> | | Background Field | Analysis Field | Ground Truth |
> | - | - | - | - |
> | z500 standard deviation | 3145.8 | 3152.6 | 3171.3 |
> | u500 standard deviation | 11.516 | 11.503 | 11.554 |
> | z500 skew | -0.1582 | -0.1586 | -0.1573 |
> | u500 skew | 0.6580 | 0.6601 | 0.6505 |
>
> > **Response to S3: Interesting but underexplored ablation insights**
>
> Thank you for highlighting this interesting finding. We also observed that LoRA yields the largest improvements for Z500, a relatively slow-evolving and spatially coherent variable. One possible reason is that Z500 reflects the large-scale circulation and tends to integrate information from multiple physical processes, making it more amenable to the structured, low-rank adaptation provided by LoRA. Conversely, the decline in performance with more finetunable parameters suggests potential overfitting or interference with pretrained representations. We will explore this further and add discussions in the revised version to strengthen the interpretation of these results.
>
> ___
>
> We appreciate your feedback once again and it helped us strengthen the overall presentation and completeness of the paper.

---

> ### Comment · Reviewer_2YwJ · 2025-08-03
>
> Thank you authors for the thorough response and updated comparisons. It's great to see that the method generalized to 4D approaches as well. Overall I'd say the rebuttal has moved in some very positive directions, but my concerns about comparisons and testing scope have not been addressed in a way that would make me comfortable increasing my rating.  I have a few more questions based on the content of the responses.
>
> > Comparison with learning-based DA methods
>
> Thanks for these detailed explanations, but it's not clear why SDA would not work in a cyclic assimilation setting - their approach is designed to work on sequences of observations. Presumably if the inclusion of a prior on $x$ is important, SDA would largely under-perform validating the hypothesis proposed in this submission.
>
> > Comparison with traditional DA methods
>
> Is there a reasonable explanation for why the relative performance of the classical DA methods is not consistent with existing literature? 3DVar appears to be quite competitive here, while historical comparisons of 3DVar vs 4D approaches like the EnKF and 4DVar have generally strongly favored 4D both on small-scale global models [1] and regional models [2]. With respect to variational methods generally, the introduction of ensemble-driven covariances has been observed to consistently produce positive or non-harmful results [3][4] while the results here are mixed.
>
> Some more information on the configurations of these methods would be helpful.  Are these results given a particular budget for each (unusually small ensembles, low numbers of observations per window in 4DVar)? If so, that's OK, but it is important to explain the limitations of the analysis. What would also be interesting here is if these restrictions are lifted, does the relative performance change? Since generating the finetuning data requires extra forward evaluations of the model, if this budget were instead spent on larger ensembles, does the proposed method still have an advantage.
>
> > More extensive testing across time
>
> Thanks for adding the additional month. Is there a reason why this can't be tested across the full test set? The seasonal analysis is extremely valuable, but it's hard to evaluate the consistency of the method from two data points.
>
> > Concerns about realism of forecast distribution
>
> Since images are unavailable in responses I'm not going to evaluate this too heavily. More informative approaches here would be a spectral analysis (magnitude/cut off curve of high frequencies) or even just images of the fields.
>
> > Calibration of the ensembles
>
> Thank you for this additional information. The results indicate that the DA approach (and the baselines) actually are not very well calibrated (post-calibration results aim to be around 1), but that is important information for the reader to know and it is not an issue that the method has limitations. The posted results are sufficient to address this concern.
>
>
> [1] Kalnay, Eugenia, et al. "4-D-Var or ensemble Kalman filter?." Tellus A: Dynamic Meteorology and Oceanography 59.5 (2007): 758-773.
> [2] Meng, Zhiyong, and Fuqing Zhang. "Tests of an ensemble Kalman filter for mesoscale and regional-scale data assimilation. Part III: Comparison with 3DVAR in a real-data case study." Monthly Weather Review 136.2 (2008): 522-540.
> [3] Buehner, M., J. Morneau, and C. Charette. "Four-dimensional ensemble-variational data assimilation for global deterministic weather prediction." Nonlinear Processes in Geophysics 20.5 (2013): 669-682.
> [4] Buehner, Mark, et al. "Intercomparison of variational data assimilation and the ensemble Kalman filter for global deterministic NWP. Part II: One-month experiments with real observations." Monthly Weather Review 138.5 (2010): 1567-1586.

---

> ### Author Response · Authors · 2025-08-07
> **Official Comment by Authors (Part 1)**
>
> Thank you for your valuable comments. We address your concerns below, point by point.
>
> ___
>
> > Comparison with learning-based DA methods
>
> Thank you for the thoughtful comment. We agree that SDA is an interesting line of work and, in principle, is designed to handle sequences of observations. However, there are practical challenges that currently limit its application in large-scale, cyclic assimilation settings such as numerical weather prediction (NWP).
>
> First, the original SDA paper was tested only on toy models (Lorenz96 [1] and a two-layer quasi-geostrophic model [2]), and its extension to high-dimensional geophysical systems presents significant engineering and training challenges. In particular, training score-based diffusion models on high-resolution atmospheric fields is much more complex than on low-dimensional toy systems. To the best of our knowledge, the most relevant recent work is Appa [3], which can be seen as an extension of SDA. However, this paper only appeared on arXiv in April 2025 and does not provide open-source code, making it difficult to reproduce under our experimental setup within the review timeframe. Furthermore, their observational setup differs significantly from ours: they simulate 11,000 observation locations based on satellite and ground stations—about 11 times more than ours—and inject artificial noise into the observations. Despite the denser observations, their reported performance is comparable to ours: RMSE on z500 stabilizes between 60–70 $\mathrm{m}^2/\mathrm{s}^2$, and on t850 around 1.2 $K$. By comparison, our LoRA-EnVar achieves z500 RMSE of 64.11 $\mathrm{m}^2/\mathrm{s}^2$ and t850 RMSE of 1.075 $K$, using fewer observations.
>
> In addition, the Appa paper only evaluates performance over a 7-day assimilation period, whereas our system is designed and tested for month-long operational settings. It remains unclear how well score-based methods like SDA or Appa would scale under such extended assimilation windows and data volumes. Thus, while we recognize the conceptual appeal of SDA, practical barriers currently limit its deployment in realistic cyclic data assimilation workflows.

---

> ### Author Response · Authors · 2025-08-07
> **Official Comment by Authors (Part 2)**
>
> > Comparison with traditional DA methods
>
> **Clarification of the experimental settings**
>
> At first, we would like to clarify that our comparison experiments are designed to be fair and consistent across all methods.
>
> 1. For *Hybrid 3DEnVar*, *Hybrid 4DEnVar*, *LoRA-3DEnVar*, and *LoRA-4DEnVar*, the ensemble size is uniformly set to 8, and the ensembles are constructed using the same time-lagged approach. For the *LETKF* method, the ensemble size is also set to 8, and the ensembles are generated using the standard LETKF procedure.
> 2. In all 3D-based methods (*3DVar*, *Hybrid 3DEnVar*, *LETKF*, *VAE-3DVar*, *LoRA-3DEnVar*), observations are sampled every 6 hours (i.e., at 0h, 6h, 12h, and 18h), with 1000 points sampled on a 721x1440 grid at each time step. In all 4D-based methods (*4DVar*, *Hybrid 4DEnVar*, *VAE-4DVar*, *LoRA-4DEnVar*), observations are distributed across finer temporal intervals within the assimilation window (i.e., at 0h, 1h, 6h, 7h, 12h, 13h, 18h, and 19h).
> 3. All AI-based methods (*VAE-3DVar*, *LoRA-3DEnVar*, *VAE-4DVar*, *LoRA-4DEnVar*) share the same core pipeline as their classical counterparts (*3DVar*, *Hybrid 3DEnVar*, *4DVar*, *Hybrid 4DEnVar*), except for the replacement of the background error covariance (typically modeled by a static or hybrid B matrix) with a learned latent representation via a VAE, optionally enhanced with LoRA-based online adaptation.
>
> Therefore, we believe the comparisons in our study are fair, self-consistent, and controlled in terms of ensemble size, observation setup, and algorithmic design.
>
> **Regarding the experimental results**
>
> Our results are broadly consistent with prior literature:
>
> 1. **4DVar vs. 3DVar**: All 4DVar-based methods (4DVar, Hybrid 4DEnVar, VAE-4DVar, LoRA-4DEnVar) outperform their 3DVar counterparts.
>
> 2. **VAE vs. traditional covariance**: VAE-based methods (VAE-3DVar, VAE-4DVar) consistently improve over their static-covariance baselines (3DVar, 4DVar).
>
> By comparing LoRA-based methods (i.e., LoRA-3DEnVar, LoRA-4DEnVar) with their traditional hybrid ensemble method counterpart (i.e., Hybrid 3DEnVar, Hybrid 4DEnVar), we find that LoRA-based methods are better, which further supports the effectiveness of our proposed method.
>
> There are, however, two aspects that may appear counterintuitive:
>  (1) The strong performance of 3DVar, and
>  (2) Mixed results from ensemble-driven covariance.
>
> **(1) Why 3DVar is so competitive?**
>
> The relative performance of DA methods depends heavily on the forecast model. In [4], EnKF only clearly outperforms 3DVar under perfect-model assumptions (e.g., SPEEDY); when using reanalysis-based observations (e.g., NCEP), this advantage diminishes significantly. Our experiments use *FengWu*, a deep learning-based forecast model with inherent model error, making our setting more comparable to the reanalysis-based case in [4]. Hence, the weaker advantage of ensemble methods over 3DVar is reasonable.
>
> Additionally, our implementation of 3DVar differs slightly from traditional configurations due to the characteristics of the AI forecast model. Specifically, while most traditional NWP systems (e.g., WRF) treat geopotential as a diagnostic variable and infer it from primitive variables [5], the AI-based FengWu model predicts geopotential directly. This distinction necessitates adaptive modifications to the 3DVar formulation [6]. In our implementation based on WRF_DA with the cv5 option, we additionally incorporated the regression coefficients between geopotential and stream function to better handle this change. Because geopotential integrates multi-variable information and is directly corrected in our formulation, the effective degrees of freedom in the assimilation are reduced, potentially improving the 3DVar performance. In contrast, traditional models must rely on nonlinear observation operators to assimilate geopotential, making the task more difficult. As a result, the 3DVar baseline in our AI-model-based setup is inherently stronger, which may help explain its unexpectedly competitive performance.

---

> ### Author Response · Authors · 2025-08-07
> **Official Comment by Authors (Part 3)**
>
> **(2) Why mixed results with ensemble covariances?**
>
> We would like to clarify that, overall, our experimental findings are consistent with the conclusions of prior literature: incorporating ensemble-driven background covariances generally improves or maintains performance across most variables. In our experiments, variables such as horizontal wind (u, v), temperature (t) exhibit positive or non-harmful improvements when transitioning from static covariance estimation to hybrid estimation. We demostrate results for other variables in the table below.
>
> | RMSE                           | 4DVar | Hybrid 4DEnVar | VAE-4DVar | LoRA-4DEnVar |
> | ------------------------------ | ----- | -------------- | --------- | ------------ |
> | u850 ($\mathrm{m}/\mathrm{s}$) | 2.305 | 2.089          | 1.940     | 1.920        |
> | v500 ($\mathrm{m}/\mathrm{s}$) | 2.787 | 2.597          | 2.496     | 2.438        |
> | v850 ($\mathrm{m}/\mathrm{s}$) | 2.273 | 2.061          | 1.888     | 1.858        |
> | t500 ($\mathrm{K}$)            | 0.721 | 0.713          | 0.739     | 0.715        |
> | t850 ($\mathrm{K}$)            | 1.138 | 1.062          | 1.054     | 1.031        |
>
> The only notable exception is the geopotential height (e.g., z500), where hybrid EnVar underperforms 3DVar. We believe this discrepancy arises from structural differences in how z is treated in our AI-based forecasting model compared to traditional NWP systems. In conventional models, geopotential is typically a *diagnostic variable*, computed from mass and thermodynamic relationships under physical constraints (e.g., hydrostatic balance). In such settings, 3DVar can effectively correct z indirectly through adjustments to other state variables or via nonlinear observation operators. However, in our AI forecasting system, geopotential is modeled as a *direct prognostic variable* without such physical coupling. This allows 3DVar to explicitly correct z via direct gradient descent on the observed variable, which can sometimes outperform ensemble-based methods when the ensemble suffers from high-dimensional sampling noise or fails to fully capture the complex flow-dependent structure of z errors.
>
> **Regarding Computational Budget**
>
> We clarify that **LoRA-3DEnVar does not incur additional model evaluations** compared to Hybrid 3DVar. Both methods use the same ensemble of 8 members, generated via time-lagged forecasts. In Hybrid 3DVar, ensemble information is incorporated through modifications to the B matrix, whereas LoRA-3DEnVar uses finetuning to achieve a similar effect.
>
> This finetuning step operates on the latent space and is computationally lightweight, requiring **less than 10 seconds** per assimilation window for an ensemble size of 8, without any extra forward model runs.

---

> ### Author Response · Authors · 2025-08-07
> **Official Comment by Authors (Part 4)**
>
> > **More extensive testing across time**
>
> Thank you for highlighting this important point. We apologize for the limited temporal coverage in the original submission. In response, we have extended our experiments to cover the entire year of 2022 and demonstrate the results in the table below. The results are generally consistent with what we have obtained for the period of one month.
>
> Results for u10 ($m/s$)
>
> | Month | 3DVar | Hybrid EnVar | VAE-Var | LoRA-EnVar |
> | ----- | ----- | ------------ | ------- | ---------- |
> | 1     | 1.541 | 1.439        | 1.262   | 1.253      |
> | 2     | 1.547 | 1.451        | 1.290   | 1.278      |
> | 3     | 1.579 | 1.494        | 1.296   | 1.272      |
> | 4     | 1.544 | 1.497        | 1.290   | 1.281      |
> | 5     | 1.631 | 1.529        | 1.367   | 1.353      |
> | 6     | 1.618 | 1.535        | 1.377   | 1.366      |
> | 7     | 1.590 | 1.516        | 1.393   | 1.387      |
> | 8     | 1.597 | 1.537        | 1.376   | 1.367      |
> | 9     | 1.648 | 1.551        | 1.349   | 1.353      |
> | 10    | 1.619 | 1.521        | 1.388   | 1.380      |
> | 11    | 1.600 | 1.508        | 1.365   | 1.322      |
> | 12    | 1.597 | 1.486        | 1.358   | 1.296      |
>
> Results for v850 ($m/s$)
>
> | Month | 3DVar | Hybrid EnVar | VAE-Var | LoRA-EnVar |
> | ----- | ----- | ------------ | ------- | ---------- |
> | 1     | 2.281 | 2.065        | 1.906   | 1.887      |
> | 2     | 2.262 | 2.060        | 1.950   | 1.946      |
> | 3     | 2.258 | 2.089        | 1.912   | 1.901      |
> | 4     | 2.274 | 2.170        | 1.950   | 1.903      |
> | 5     | 2.386 | 2.194        | 2.034   | 1.999      |
> | 6     | 2.397 | 2.194        | 2.053   | 2.018      |
> | 7     | 2.398 | 2.219        | 2.064   | 2.035      |
> | 8     | 2.389 | 2.220        | 2.052   | 2.055      |
> | 9     | 2.433 | 2.234        | 2.035   | 2.042      |
> | 10    | 2.354 | 2.159        | 2.034   | 2.031      |
> | 11    | 2.298 | 2.098        | 1.998   | 1.944      |
> | 12    | 2.269 | 2.084        | 1.967   | 1.946      |
>
> Results for t850 ($K$)
>
> | Month | 3DVar | Hybrid EnVar | VAE-Var | LoRA-EnVar |
> | ----- | ----- | ------------ | ------- | ---------- |
> | 1     | 1.145 | 1.070        | 1.094   | 1.060      |
> | 2     | 1.108 | 1.047        | 1.150   | 1.115      |
> | 3     | 1.097 | 1.051        | 1.165   | 1.128      |
> | 4     | 1.110 | 1.104        | 1.170   | 1.094      |
> | 5     | 1.136 | 1.092        | 1.165   | 1.119      |
> | 6     | 1.145 | 1.093        | 1.157   | 1.113      |
> | 7     | 1.128 | 1.091        | 1.192   | 1.141      |
> | 8     | 1.145 | 1.107        | 1.151   | 1.105      |
> | 9     | 1.140 | 1.091        | 1.119   | 1.095      |
> | 10    | 1.163 | 1.104        | 1.154   | 1.098      |
> | 11    | 1.165 | 1.119        | 1.182   | 1.104      |
> | 12    | 1.170 | 1.117        | 1.201   | 1.104      |
>
> Results for mslp ($Pa$)
>
> | Month | 3DVar | Hybrid EnVar | VAE-Var | LoRA-EnVar |
> | ----- | ----- | ------------ | ------- | ---------- |
> | 1     | 250.6 | 190.0        | 284.1   | 171.2      |
> | 2     | 212.7 | 182.0        | 290.2   | 221.3      |
> | 3     | 202.0 | 192.2        | 254.0   | 173.7      |
> | 4     | 198.4 | 204.7        | 291.2   | 191.7      |
> | 5     | 188.3 | 216.6        | 290.5   | 212.0      |
> | 6     | 217.2 | 207.3        | 374.0   | 220.8      |
> | 7     | 214.4 | 191.2        | 430.0   | 231.4      |
> | 8     | 209.5 | 187.3        | 460.5   | 252.6      |
> | 9     | 234.1 | 219.0        | 403.4   | 241.4      |
> | 10    | 242.8 | 250.4        | 419.9   | 249.8      |
> | 11    | 256.1 | 256.9        | 428.0   | 245.4      |
> | 12    | 253.9 | 250.1        | 433.1   | 243.9      |

---

> ### Author Response · Authors · 2025-08-07
> **Official Comment by Authors (Part 5)**
>
> > **Concerns about realism of forecast distribution**
>
> Thank you for your understanding. We will include visualizations of both the analysis fields and the corresponding spectral analysis in the revised version to better illustrate the high-frequency behavior.
>
> > **Calibration of the ensembles**
>
> Thank you for the helpful clarification. We appreciate the acknowledgement that the posted results sufficiently address the concern. While improving calibration was not the primary focus of this work, we agree that reporting these metrics provides valuable context. We will clarify this limitation in the revised version to ensure transparency for the reader.
>
> ---
>
> **References**
>
> [1] Rozet, F., & Louppe, G. (2023). Score-based data assimilation. *Advances in Neural Information Processing Systems*, *36*, 40521-40541.
>
> [2] Rozet, F., & Louppe, G. (2023). Score-based data assimilation for a two-layer quasi-geostrophic model. *arXiv preprint arXiv:2310.01853*.
>
> [3] Andry, G., Rozet, F., Lewin, S., Rochman, O., Mangeleer, V., Pirlet, M., ... & Louppe, G. (2025). Appa: Bending weather dynamics with latent diffusion models for global data assimilation. *arXiv preprint arXiv:2504.18720*.
>
> [4] Kalnay, E., Li, H., Miyoshi, T., Yang, S. C., & Ballabrera-Poy, J. (2007). 4-D-Var or ensemble Kalman filter?. *Tellus A: Dynamic Meteorology and Oceanography*, *59*(5), 758-773.
>
> [5] Meng, Z., & Zhang, F. (2008). Tests of an ensemble Kalman filter for mesoscale and regional-scale data assimilation. Part III: Comparison with 3DVAR in a real-data case study. *Monthly Weather Review*, *136*(2), 522-540.
>
> [6] Adrian, M., Sanz-Alonso, D., & Willett, R. (2025). Data assimilation with machine learning surrogate models: A case study with FourCastNet. *Artificial Intelligence for the Earth Systems*, *4*(3), e240050.

---

> > ### Comment · Reviewer_2YwJ · 2025-08-08
> >
> > I thank the authors for the discussion, but while my concern about limited sample sizes has been addressed, the discussion has deepened my concern over the quality of the baseline comparisons. The paper has some very interesting ideas and the core concept seems well-founded, but the unexpected relative performance and the poor calibration of the baselines suggest weak implementations which erodes trust in the results.
> >
> > From a reader's perspective, the most plausible explanation for the unexpected overperformance of 3D methods is the observation generation strategy which samples noiseless observations taking place at the same time as assimilation steps. This avoids many of the complications of operational DA, but also renders this an unrealistic setting as handling temporally separated observations is why 3D methods are so uncommon. This weakness of the paper could be mitigated by a thorough discussion of the limitations, but in the present framing, the paper claims this to be a realistic operational setting.
> >
> > Other likely factors include smaller ensemble sizes. Again, strong performance at small ensemble sizes is valuable, but it is also true that the regime here is significantly smaller than typical so it needs to be explicit how this setting is unusual to both illustrate why this is an advantage and whether these advantages go away in larger ensemble settings. Given it is unlikely that ensemble sizes will shrink significantly for probabilistic forecast reasons, this is important information to know.
> >
> > Ultimately, the paper would benefit significantly from more careful implementation of the baselines as well a more thorough analysis of the non-trivial experimental limitations. Given these issues, I cannot adjust my score despite the strides made in other areas. Below, I'll outline some points of discussion that contributed to my unease with the current submission:
> >
> > >  In [4], EnKF only clearly outperforms 3DVar under perfect-model assumptions (e.g., SPEEDY); when using reanalysis-based observations (e.g., NCEP), this advantage diminishes significantly.
> >
> > While not inaccurate, this statement is misleading as it ignores what the numbers actually show. The reference shows that the methods go from completely non-competitive (75% improvement using EnKF) to relatively modest improvements. Specifically, the section on NCEP reanalysis shows the RMSE improvement on z500 is only 25% which is still quite large, though other pressure levels show smaller advantages. Bias-correction of the weak dynamics model increased this to 50% and made it more consistent across pressure levels.
> >
> > The rebuttal, using a state of the art dynamics model specifically trained to reproduce ERA5, is showing that 3DVar is performing 30% better on this field. This is a significant deviation from previous analysis in atmospheric settings which is concerning.
> >
> > > SDA Discussion
> >
> > I fully agree that lack of open source software capable of performing the task is a valid argument for excluding a comparison. My concern here is the use of the claim that the comparison is excluded because it doesn't scale. There doesn't seem to be any grounding for this in the form of references to prior attempts or complexity analysis, and there does appear to be recent (out of scope) work that did employ this approach.
> >
> > Given the primary experiments do not take place in a realistic setting, it would likely help the paper to include smaller scale tests on better understood systems. This might allow for more thorough baselines against ML approaches. If the advantages are clear from a thorough comparison against ML baselines, then it is no longer as vital to have well-tuned classical baselines since the claimed contributions could be adjusted to specify that the method is only shown to outperform existing ML approaches which would still be significant.
> >
> > Additionally, the comments on the out of scope paper highlight some of why I am worried about the quality of the comparisons. Similar to the previous bullet, these claims again are again obscuring trade-offs that should be openly discussed. The authors claim that performance is similar between the submission with 11x fewer observations. However, the reality is significantly trickier. In the Appa paper, it is not clear what the breakdown of "station" and "satellite" observations are, but the station observations referred to in the reply observe only surface variables. This is a highly clustered sampling strategy resulting in approx 6x11k observed scalar measurements with added noise. In this submission, there are only 1000 sampling stations, but each assimilates full field observations for 69x1000 observed scalar values and the sampling strategy appears to be uniform and without noise. Which provides more information is actually unclear.

---

> > > ### Author Response · Authors · 2025-08-09
> > >
> > > Dear Reviewer,
> > >
> > > Thank you for your thorough evaluation and for sharing these detailed concerns. We appreciate your constructive feedback on baseline calibration, observation settings, and experimental realism. We will incorporate a more careful discussion of these limitations, refine our baseline implementations, and expand the evaluation to include larger ensembles and more realistic observation scenarios in future revisions. Your comments will be highly valuable for strengthening the next version of this work.
> > >
> > > Best regards,
> > >
> > > The Authors of Submission 8287

---

### Official Review · Reviewer_Y4Yp · 2025-07-02

**Clarity:** 3
**Significance:** 3
**Originality:** 3
**Rating:** 5
**Confidence:** 3

**Summary:**

The authors use finetuned LoRA modules and combine them with climate error models (as in VAE-VAR) to conduct data assimilation. They do this by first training a VAE decoder on historical error values and then creating an ensemble of predictions for each cycle and finetuning the LoRA matrices for the flow-dependent ensemble perturbations.

**Questions:**

Why is are only the query LoRA integrated, as compared to also the K and the V?

Is there a link between the LoRA rank and an assumed effective ensemble size or background covariance rank?

**Ethical Concerns:**

["NO or VERY MINOR ethics concerns only"]

**Final Justification:**

I believe that this work is a good contribution towards the field of data assimilation, and I will argue for acceptance. The experiments seem to be robust on ERA5. However, since its improvements over the standard VAE-Var approach seems to be somewhat incremental, its impact is slightly limited to be considered for a strong acceptance.

**Limitations:**

Yes.

**Paper Formatting Concerns:**

None.

**Quality:**

3

**Strengths And Weaknesses:**

Strengths

The model is efficient in terms of trainable parameters at test time which is only 0.1 % of the model, which is practical for finetuning with limited GPU capacity when running in an online fashion. This allows one to adapt it to more complex transformer architectures for the VAE without much computational overhead.

The authors do not assume a Gaussian background which allows for more flexible likelihood functions compared to standard data assimilation models which assume Gaussian backgrounds.

The authors test the model at 0.25 degree global resolution, and they found that LoRA with rank 2 was sufficient to capture the detailed dataset, so method scales without exponential parameter growth. 0.25 degree forecasting is the de-facto standard for MLWP today.

The model also seems robust to different ensemble sizes, observation densities, and variables. The authors have done quite a few ablation experiments while adjusting these parameters.

The integration of LoRA into standard diffusion model transformers for image generation has been explored before, but it has not been applied to the weather prediction and assimilation community. This paper presents the first foray into incorporating such a step, and also offers avenues of improvements for future data assimilation techniques which are based on a VAE-Var style of data assimilation.

LORA EnVar seems to scale well with observation sparsity in comparison to the baseline methods. However, the results are not a significant boost over strong baselines such as VAE-Var.

Weaknesses

The model relies on a Swin-V2 backbone tailored to FengWu; portability to other forecast models is not clear in terms of how it goes.
Although LoRA updates are light, the decoder is still a large transformer that must run in each inner-loop gradient evaluation.

Summary

The paper is well-structured and clear, and offers significant novelty in terms of introducing LoRA to data assimilation with MLWP, and opens up future avenues of improvement for VAE-Var style data assimilation approaches. The results seem like a modest improvement upon existing data assimilation techniques like VAE-Var. As a result, I vote accept for this paper.

---

> ### Author Rebuttal · Authors · 2025-07-27
>
> Thank you very much for your constructive and detailed comments. We greatly appreciate your positive assessment of the methodology, novelty, and overall presentation of our work.
>
> ___
>
> > **Response to Q1: Selective LoRA integration on queries**
>
> Thank you for the insightful question regarding the selective integration of LoRA modules into the query projections only.
>
> To address this, we conducted additional ablation experiments where LoRA was applied to different combinations of the query (Q), key (K), and value (V) projections. Specifically, we tested:
> (1) finetuning Q only (the default in our main results),
> (2) finetuning both Q and K,
> (3) finetuning both Q and V, and
> (4) finetuning Q, K, and V simultaneously.
>
> The results over the first 30 days of January are summarized in the tables below. For each 3-day period, we computed the average RMSE across twelve 6-hour assimilation cycles. These experiments were conducted for two representative variables: z500 and u500. As shown, finetuning Q alone already achieves strong performance, and while incorporating additional LoRA modules into K and/or V leads to minor improvements in some periods, the relative performance fluctuates across different 3-day windows. Overall, the average gains are marginal and often fall within the variability observed across repeated runs.
>
> Importantly, extending LoRA to K or V increases the number of trainable parameters by $2\times$ or $3\times$, which goes against our design goal of efficient online adaptation with minimal computational overhead. Thus, we chose to finetune only Q in our main experiments.
>
> | z500 RMSE ($\mathrm{m^2}/\mathrm{s^2}$) | Finetune Q | Finetune Q, K | Finetune Q, V | Finetune Q, K, V |
> | --------------------------------------- | ---------- | ------------- | ------------- | ---------------- |
> | Day 1-3                                 | 56.79      | 56.46         | 56.25         | 54.62            |
> | Day 4-6                                 | 59.77      | 60.46         | 66.67         | 65.49            |
> | Day 7-9                                 | 61.24      | 61.47         | 62.10         | 60.80            |
> | Day 10-12                               | 65.34      | 64.95         | 65.43         | 64.72            |
> | Day 13-15                               | 65.97      | 66.41         | 64.57         | 64.79            |
> | Day 16-18                               | 63.85      | 65.14         | 63.59         | 65.34            |
> | Day 19-21                               | 67.88      | 68.72         | 64.95         | 65.30            |
> | Day 22-24                               | 65.17      | 66.33         | 69.48         | 71.20            |
> | Day 25-27                               | 67.31      | 69.27         | 62.24         | 65.58            |
> | Day 28-30                               | 67.92      | 71.07         | 65.00         | 66.90            |
> | Average                                 | 64.12      | 65.03         | 64.03         | 64.47            |
>
> | u500 RMSE ($\mathrm{m}/\mathrm{s}$) | Finetune Q | Finetune Q, K | Finetune Q, V | Finetune Q, K, V |
> | ----------------------------------- | ---------- | ------------- | ------------- | ---------------- |
> | Day 1-3                             | 2.403      | 2.398         | 2.384         | 2.384            |
> | Day 4-6                             | 2.599      | 2.595         | 2.582         | 2.567            |
> | Day 7-9                             | 2.605      | 2.592         | 2.598         | 2.551            |
> | Day 10-12                           | 2.701      | 2.675         | 2.722         | 2.659            |
> | Day 13-15                           | 2.554      | 2.584         | 2.571         | 2.547            |
> | Day 16-18                           | 2.634      | 2.630         | 2.612         | 2.589            |
> | Day 19-21                           | 2.781      | 2.741         | 2.753         | 2.730            |
> | Day 22-24                           | 2.683      | 2.656         | 2.700         | 2.693            |
> | Day 25-27                           | 2.657      | 2.612         | 2.649         | 2.608            |
> | Day 28-30                           | 2.674      | 2.706         | 2.672         | 2.661            |
> | Average                             | 2.629      | 2.619         | 2.624         | 2.599            |
>
> These findings are also consistent with the original LoRA paper [1], which reports that finetuning Q alone often suffices, and that additional gains from finetuning K and V are typically small. This reinforces our decision to prioritize efficiency by limiting updates to the query projections only.
>
> > **Response to Q2: Relationship between the LoRA rank and an effective ensemble size**
>
> Thank you for the insightful question regarding the relationship between LoRA rank and ensemble size or background covariance rank.
>
> While both the ensemble size in traditional data assimilation and LoRA rank in our framework reflect constraints on the representational capacity of uncertainty, they operate in fundamentally different ways. In traditional ensemble-based methods, ensemble size determines the number of samples used to estimate the flow-dependent structure of the background error. A small ensemble size can lead to an under-represented covariance with artificially low rank, failing to capture key dynamical modes. In general, assuming sufficient computational resources are available, larger ensembles lead to better performance, as they provide more accurate and higher-rank estimates of the background error.
>
> In contrast, LoRA rank does not directly correspond to ensemble size, nor does it determine the rank of any explicit background covariance matrix. Instead, it controls the maximum number of directions in parameter space along which dynamical corrections can be introduced into a climatologically trained model. If the LoRA rank is too low, even a large ensemble cannot be fully leveraged, as the model lacks sufficient degrees of freedom to incorporate flow-dependent corrections. Conversely, an overly high rank may introduce redundant parameters with little benefit. In our experiments, a LoRA rank of 2  achieved a optimal balance between expressiveness and efficiency. Moreover, in LoRA-EnVar, flow-dependent information is incorporated via online learning, enabling the model to extract meaningful dynamical information even with a small ensemble, as long as the LoRA rank is sufficiently expressive.
>
> Exploring a more explicit theoretical connection between LoRA rank, ensemble size, and the effective background error structure remains an interesting direction for future work.
>
> > **Response to W1: Portability to backbones beyond Swin-V2**
>
> We thank the reviewer for raising the question of portability. Our current implementation of LoRA-EnVar is based on the Swin-V2, but the approach is in principle easily adaptable to other Transformer-based architectures. This flexibility is because LoRA modules operate over standard linear projections, such as query/key/value matrices, which are widely shared across Transformer variants.
>
> For models like GraphCast, which are based on graph neural networks (GNNs) and do not contain Transformer-style attention mechanisms, a direct application of LoRA-EnVar is not straightforward. Such architectures lack the projection layers that LoRA typically targets. Adapting our framework to GNNs would require non-trivial modifications, such as developing alternative fine-tuning mechanisms tailored for message-passing networks. We consider this a promising direction for future work.
>
> > **Response to W2: Inner-loop gradient updates requiring full transformer backpropagation**
>
> We appreciate the reviewer’s insightful observation. Indeed, our method requires running the Swin-V2 decoder in each inner-loop gradient evaluation during assimilation, reflecting the inherent computational cost of variational approaches that rely on gradient optimization. However, by updating only the lightweight LoRA modules, we significantly reduce the number of **trainable parameters**. While the full decoder remains part of the backward graph to allow gradient flow into the LoRA parameters, all backbone weights remain frozen, eliminating gradient storage for the backbone itself. This approach reduces memory usage and simplifies optimization, even though backpropagation still traverses the full network. In future work, we plan to explore further optimizations, such as updating only a subset of decoder layers or employing lighter-weight backbone architectures.
>
> > **Response to S6 (last sentence): Comparison with other baselines, such as VAE-Var**
>
> We appreciate the reviewer’s comment. Indeed, even in traditional data assimilation methods, the gain from incorporating flow-dependent (dynamical) error structures is often incremental, particularly on some variables, when compared to strong climatological baselines (e.g., Hybrid 3DEnVar vs. 3DVar, as shown in Figure 3). Our goal is to introduce such capabilities in a lightweight and extensible way, and we believe LoRA-EnVar achieves this with minimal overhead.
>
> ___
>
> We appreciate your feedback once again and it helped us strengthen the overall presentation and completeness of the paper.
>
> **References:**
>
> [1] Hu, E. J., Shen, Y., Wallis, P., Allen-Zhu, Z., Li, Y., Wang, S., ... & Chen, W. (2022). Lora: Low-rank adaptation of large language models. *ICLR*, *1*(2), 3.

---

> > ### Comment · Reviewer_Y4Yp · 2025-08-03
> >
> > Thanks for the detailed response to my questions! Reading the replies and the other reviewers' comments, I would like to maintain my position that this work is a good contribution towards the field of data assimilation. However, since its improvements over the standard VAE-Var approach seems to be somewhat incremental, its impact is slightly limited to be considered for a strong acceptance.

---

> > > ### Author Response · Authors · 2025-08-03
> > >
> > > Thank you for the positive support and thoughtful assessment. We appreciate your encouraging feedback.

---

### Comment · Area_Chair_EQNp · 2025-08-03

Dear authors and reviewers,

First of all, thank you all for your efforts so far. The author-reviewer discussion period will end on August 6.

@Authors: If not done already, please answer all questions raised by the reviewers. Remain factual, short and concise in your responses, and make sure to address all points raised.

@Reviewers: Read the authors' responses and further discuss the paper with the authors if necessary. In particular, if the concerns you raised have been addressed, take the opportunity to update your review and score accordingly. If some concerns remain, or if you share concerns raised by other reviewers, please make sure to clearly state them in your review. In this case, consider updating your review accordingly (positively or negatively). You can also maintain your review as is, if you feel that the authors' responses did not address your concerns.

I will reach out to you again during the reviewer-AC discussion period (August 7 to August 13) to finalize the reviews and scores.

The AC

---

> ### Comment · Area_Chair_EQNp · 2025-08-08
>
> Dear reviewers,
>
> The reviewers-authors discussion phase will end in less than 24 hours.
>
> If not done already, make sure to submit the "Mandatory Acknowledgement" that confirms that you have read the reviews, participated in the discussion, and provided final feedback in the "Final justification" text box.
>
> Be mindful of the time and efforts the authors have invested in answering your questions and at least acknowledge their responses. Make sure to provide a fair and scientifically grounded review and score. If you have changed your mind about the paper, please update your review and score accordingly. If you have not changed your mind, please provide a clear and sound justification for your final review and score.
>
> Best regards,
> The AC

---

### Author Response · Authors · 2025-08-07
**General Response**

**Dear Reviewers, ACs, and SACs,**

We sincerely thank you for your valuable time, thoughtful feedback, and engaging discussions throughout the review process!

---

We are particularly encouraged by the recognition of our contributions across multiple dimensions:

- Reviewer `pe2M` praised the *"vast efficiency improvement"* achieved in a *"vitally important application"*.
- Reviewer `scAN` found the paper *"clearly written"*, and described our method as *"both intuitive and powerful"*.
- Reviewer `Zgna` highlighted the *"elegant and interesting approach"*, as well as the *"strong experimental section"* with *"all components justified"*.
- Reviewer `2YwJ` emphasized the *"promising performance at smaller ensemble sizes"* and appreciated the *"interesting extensive ablations"*.
- Reviewer `Y4Yp` acknowledged our method’s *"efficiency"*, *"flexible likelihood functions"*, *"robustness"*, and the *"first foray incorporating LoRA into data assimilation"*.

---

In response to your constructive suggestions, we have made the following clarifications and improvements:

- **More comprehensive experimental results**
  - As suggested by Reviewer `2YwJ`, we have conducted additional one-year-long assimilation experiments to evaluate our method over extended time periods.
  - As suggested by Reviewers `2YwJ` and `scAN`, we have included the 4DVar extension of LoRA-EnVar and compared it with traditional 4DVar algorithms.
  - As suggested by Reviewer `Y4Yp`, we have added additional ablation studies on the KQV LoRA modules.
  - As suggested by Reviewer `pe2M`, we have included an overall metrics comparison across different methods.
  - As suggested by Reviewer `2YwJ`, we have added experiments on the calibration of the background ensemble and comparisons of the resulting distributions.
- **Computational cost**
  - As suggested by Reviewers `2YwJ` and `scAN`, we have clarified that a primary contribution of LoRA lies in its significant reduction of parameter count and storage requirements.
- **Relation to other AI methods**
  - As suggested by Reviewers `2YwJ` and `scAN`, we have emphasized the advantages of our method over other AI approaches, particularly in its ability to handle high-dimensional, real-world weather prediction tasks and to accommodate both static background error modeling and dynamic error structures.
- **Flexibility with regard to ensemble generation**
  - As suggested by Reviewer `Zgna`, we have clarified that our method is robust to different ensemble generation strategies.

---

We would like to re-emphasize the **key contributions** of our work:

- We introduce the **first hybrid deep generative framework** that unifies VAE-based climatological background modeling with LoRA-based flow-dependent adaptation for data assimilation.
- We enable **efficient online low-rank finetuning** within assimilation cycles, allowing for dynamic, non-Gaussian background error updates while reducing the number of trainable parameters by three orders of magnitude.
- Our framework is **agnostic to the observation operator** (supporting various observation types and both 3DVar/4DVar settings) and is **robust to different ensemble generation strategies**.
- We provide **comprehensive ablation studies** and evaluations on **real-world**, **high-dimensional** forecasting systems, demonstrating the effectiveness and versatility of our method.

---

With **two days** remaining in the **Author-Reviewer Discussion** phase, we welcome any further feedback and are happy to continue the discussion.

Thank you once again for your thoughtful reviews and kind support!

Warmest regards,

The Authors of Submission 8287

---

### Decision · Program_Chairs · 2025-09-17

**Decision:**

Accept (poster)

**Comment:**

The average rating is 4.6, with almost all reviewers recommending acceptance (4, 5, 6, 5, 3). The discussion has been constructive and improvements have been made to the paper, to the satisfaction of the reviewers. Reviewer 2Ywj (3) has raised concerns about the realism of the observational model considered in the experiments and the choice of baselines. Given the overall positive reviews, I recommend acceptance. However, I strongly encourage the authors to follow Reviewer 2Ywj's suggestions and discuss the realism or artificiality of the experimental setup and adjust claims accordingly. New results presented during the discussion are helpful and should be included in the final version of the paper.

Recommendation: acceptance.